# Structural rearrangement of amyloid-β upon inhibitor binding suppresses formation of Alzheimer's disease related oligomers

Tobias Lieblein[1†], Rene Zangl[1†], Janosch Martin[1], Jan Hoffmann[1], Marie J Hutchison[2], Tina Stark[2], Elke Stirnal[2], Thomas Schrader[3], Harald Schwalbe[2], Nina Morgner[1*]

[1]JW Goethe-University, Institute of Physical and Theoretical Chemistry, Frankfurt, Germany; [2]JW Goethe-University, Institute for Organic Chemistry and Chemical Biology and Center for Biomolecular Magnetic Resonance, Frankfurt am Main, Germany; [3]University of Duisburg-Essen, Institute of Organic Chemistry, Essen, Germany

**Abstract** The formation of oligomers of the amyloid-β peptide plays a key role in the onset of Alzheimer's disease. We describe herein the investigation of disease-relevant small amyloid-β oligomers by mass spectrometry and ion mobility spectrometry, revealing functionally relevant structural attributes. In particular, we can show that amyloid-β oligomers develop in two distinct arrangements leading to either neurotoxic oligomers and fibrils or non-toxic amorphous aggregates. Comprehending the key-attributes responsible for those pathways on a molecular level is a pre-requisite to specifically target the peptide's tertiary structure with the aim to promote the emergence of non-toxic aggregates. Here, we show for two fibril inhibiting ligands, an ionic molecular tweezer and a hydrophobic peptide that despite their different interaction mechanisms, the suppression of the fibril pathway can be deduced from the disappearance of the corresponding structure of the first amyloid-β oligomers.

*For correspondence:
morgner@chemie.uni-frankfurt.de

†These authors contributed equally to this work

Competing interests: The authors declare that no competing interests exist.

## Introduction

Dementia is a widespread condition in the western civilization with Alzheimer's disease (AD) being the most common form (*Folch et al., 2018*; *Hamley, 2012*; *Kumar and Hamilton, 2017*). Up to now, no effective treatment of AD has been developed, and substantial efforts are underway to develop drugs that effectively inhibit AD pathogenesis. The cause of onset and progression of AD is unknown, but evidence points to the increased formation of oligomers of the neurotoxic amyloid-β peptide ($A\beta_{42}$) playing a major role in the development of AD (*Hamley, 2012*; *Anand et al., 2014*; *Bu et al., 2016*; *Defelice and Ferreira, 2002*; *Gilbert, 2014*; *Lee et al., 2017*; *Zhao et al., 2012*). The soluble $A\beta_{42}$ oligomers aggregate into fibrils and the detrimental effects of AD seem to be caused by neurotoxicity of the oligomers (*Habchi et al., 2017*; *Dobson, 2003*). Investigation of the oligomerization pathway to identify oligomeric states and interactions stabilizing these states is therefore imperative. When isolated in solution, $A\beta_{42}$ shows characteristic patterns of an unfolded polypeptide chain lacking persistent tertiary structure (*Habchi et al., 2016*; *Chiti et al., 2003*). Depending on concentration, temperature, and homogeneity of the starting material, the polypeptide chain rapidly forms fibrils. The significance of the exact peptide structure for the aggregation process was suggested by molecular dynamic simulations (MD) (*Barz et al., 2018*). Nevertheless, structure determination of the fibrillar state of $A\beta_{42}$ has long been challenging (*Miller et al., 2010*)

and only recently structures for $A\beta_{42}$ fibrils were determined using solid-state nuclear magnetic resonance (NMR) and cryo-electron-microscopy (EM) (*Colvin et al., 2016*; *Gremer et al., 2017*; *Wälti et al., 2016*; *Xiao et al., 2015*). The analysis of $A\beta_{42}$ structure by solid-state NMR and by cryo-EM consistently revealed an S shaped conformation formed by two branches of $A\beta_{42}$ as the basis structure within the fibril. The first branch is stabilized via hydrophobic interactions between amino acids L17 to I32 (*Colvin et al., 2016*; *Gremer et al., 2017*; *Wälti et al., 2016*; *Xiao et al., 2015*), including the hydrophobic KLVFF region (residues 16–20). The second branch is stabilized via an ionic interaction between the K28 sidechain and the C-terminal A42 (*Colvin et al., 2016*; *Wälti et al., 2016*; *Xiao et al., 2015*). Oligomerization of monomeric $A\beta_{42}$ is conceivable as a parallel stacking in the axial direction of fibril growth. However, recently published fibril structures show fibrillization to occur not via a stacking of a monomeric $A\beta_{42}$ base (MB), but with a dimeric $A\beta_{42}$ base (DB) as the basic module for fibrillary stacking. DB consists of two S shaped $A\beta_{42}$ molecules ordered in a C2 symmetric ying-yang-like fashion (*Colvin et al., 2016*; *Gremer et al., 2017*; *Wälti et al., 2016*). This DB nucleation of $A\beta_{42}$ leads typically to fibrils which are known as the on-pathway aggregates. At the same time, $A\beta_{42}$ aggregation might deviate from this oligomerization following a so-called off-pathway aggregation. The aggregates formed via the off-pathway are amorphous and do not show toxicity (*Hamley, 2012*; *Lee et al., 2017*; *Jiang et al., 2012*; *Kłoniecki et al., 2011*; *Sitkiewicz et al., 2014*; *Taylor et al., 2010*; *Woods et al., 2013*). Until today, several mass spectrometry (MS) studies investigated $A\beta_{42}$ oligomerization (*Bernstein et al., 2009*; *Pujol-Pina et al., 2015*). The combination of MS with ion mobility spectrometry (IMS) has shown differences stemming from different $A\beta_{42}$ isoforms (*Bernstein et al., 2005*; *Soper et al., 2013*) as well as ligand binding (*Young et al., 2015*; *Zheng et al., 2015*; *Beck et al., 2015*). However, no structural information deduced from IMS measurements has yet allowed deciphering the fundamental mechanism of fibril growth and no structural basis for interfering with potential on- and off pathway oligomer formation by inhibitors of low molecular weight could be elucidated. Thus, we applied native MS combined with IMS to investigate the formation of the first $A\beta_{42}$ oligomers. While oligomers of $A\beta_{42}$ with same size but different arrangements do not differ in mass, they do in their shapes which makes it possible to separate the different species in an electrospray ionization (ESI) IM experiment (*Hoffmann et al., 2017*; *Konijnenberg et al., 2013*; *Young et al., 2014*). This allowed us to delineate the structural differences between peptides aggregating via different oligomerization pathways to determine attributes relevant for the toxicity of $A\beta_{42}$. Further, for two $A\beta_{42}$ ligands OR2 (*Austen et al., 2008*) and CLR01 (*Sinha et al., 2011*) with different interaction modes, if binding to $A\beta_{42}$, we can show structural and kinetic differences in inhibition of the same on-pathway oligomerization and deduce a structural basis for their mode of action. Comprehension of the molecular mechanisms controlling the on- or off-pathway aggregation of $A\beta_{42}$ is pre-requisite for the design of pharmaceutic agents suppressing $A\beta_{42}$ oligomerization and thereby neurotoxicity.

## Results

### Every $A\beta_{42}$ oligomer has two arrangements

*Figure 1A* depicts an ion mobility (IM) driftscope plot of $A\beta_{42}$. Oligomers from monomer to nonamer with several charge states in different intensities can be observed. The IMS driftscope spectrum reveals more than one drift time for every m/z value, either arising from different tertiary or quaternary structures for each of the oligomers or an overlap of the charge distributions of different oligomeric species. In such cases where the origin of the measured peaks is not directly clear, the isotopic pattern of the m/z signals can be used to unambiguously determine the mass of the correlating species. While the isotopic patterns show that some overlap occurs, there is clear evidence that all oligomers adopt several conformations. The insets in *Figure 1A* show an example for both cases. Inset (ii) depicts for the MS peak of m/z = 2258 several species with different drift times. These show different isotope distributions, by which they can be assigned to the respective oligomers, showing that this mass peak contains signal of several overlapping oligomers. In contrast, inset (i) shows for the MS peak at m/z = 1806 an identical isotope distribution for all three appearing drift time signals, indicating the existence of only one oligomer – the dimer (5-times charged) with three conformations. The two smaller of these three dimer conformations represent structures present in solution (*Kaltashov and Eyles, 2002*; *Ruotolo and Robinson, 2006*). The signal with the lowest drift

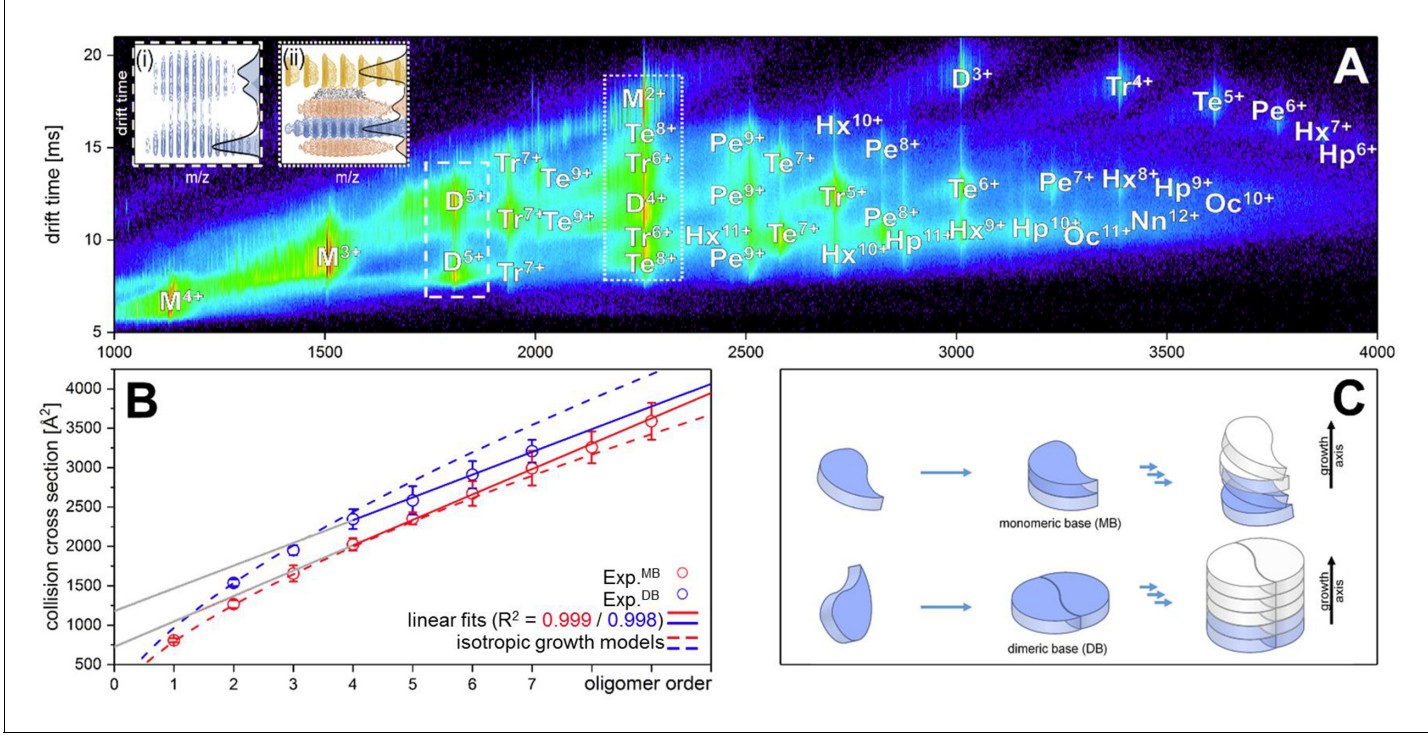

**Figure 1.** Aβ$_{42}$forms oligomers with two conformations. (**A**) Shows the driftscope spectrum of aggregated Aβ$_{42}$ polypeptide. Signals corresponding to oligomers up to nonamer in several charge states and several conformations can be seen (detailed description of the peak assignment can be found in *Appendix 1—figure 1A*). The intensity is decoded logarithmically in a heat map where red denotes a high, and blue a low intensity. The observable oligomers are denoted as follows: M = monomer, D = dimer, Tr = trimer, Te = tetramer, Pe = pentamer, Hx = hexamer, Hp = heptamer, Oc = octamer, Nn = nonamer. The two insets show driftscope zooms of the indicated m/z areas. (i) Shows the driftscope of the 5-times charged Aβ$_{42}$ dimer at m/z = 1806. The identical isotope distribution of the three peaks unambiguously indicate the presence of three different conformations for the same oligomeric state (blue). The different isotopic distributions in (ii) show that the mass peak of m/z = 2258 consists of overlapping species of several oligomers (the dominating ones are depicted in yellow, orange, blue). For the experimentally determined oligomers in A, the CCS were calculated. (**B**) depicts the CCS for all oligomers which could be assigned to solution conformations of the respective oligomer (red and blue circles). For comparison, the lines indicate theoretically calculated CCS following an isotropic growth model (dotted lines) or linear growth model (solid lines) (*Bleiholder et al., 2011*). (**C**) depicts the proposed process during the first self assembly steps of Aβ$_{42}$ peptides. The first two monomers can form a planar dimer, which is the base (DB) for further addition of Aβ$_{42}$ monomers. If no planar dimer is formed, the Aβ$_{42}$ monomers stack axially onto the monomeric Aβ$_{42}$ base (MB). This results in two morphologies: A single stranded aggregate with a MB conformation, and a zipper like structure with a DB arrangement for fibril growth.

time decreases in intensity with a higher collision energy (CE), while the signal with the highest drift time increases (*Figure 2A*). This is due to collision induced unfolding (CIU) of the Aβ$_{42}$ ion in the gas phase into the larger conformation. The same observation holds true for all oligomers. For every oligomer, we observe two differently structured oligomeric states and for higher lab frame energies (higher charge states and/or higher collision energies), two additional extended structures as the result of unfolding of the two compact species (see as well *Figure 2C*). No dominating larger oligomer, such as a 'magic' hexamer, as suggested previously, was found (further discussion in Fig. *Appendix 1—figure 1* and *2*; *Bernstein et al., 2005*). To better understand the oligomerization process, we used the drift times of the structured oligomeric states to calculate their collision cross-section (CCS) (*Laszlo and Bush, 2017*; *Stow et al., 2017*; *Warnke et al., 2017*). Experimental CCS can be compared to theoretical growth models and to theoretically calculated CCS of structural models. As it is under debate whether low mass Aβ$_{42}$ oligomers already possess a stable structure, rather than having predominantly unstructured features, (*Kumar and Hamilton, 2017*; *Colvin et al., 2016*; *Abedini and Raleigh, 2009*; *Rauk, 2009*) we first calculated CCS fitting isotropic and linear growth models as published by Bleiholder, et al. (*Figure 1B*; *Bleiholder et al., 2011*). Proteins, aggregating in an unstructured fashion would resemble an isotropic growth pattern while fibrillary growth would match a linear fit. We calculated growth models for comparison with both sets of experimentally

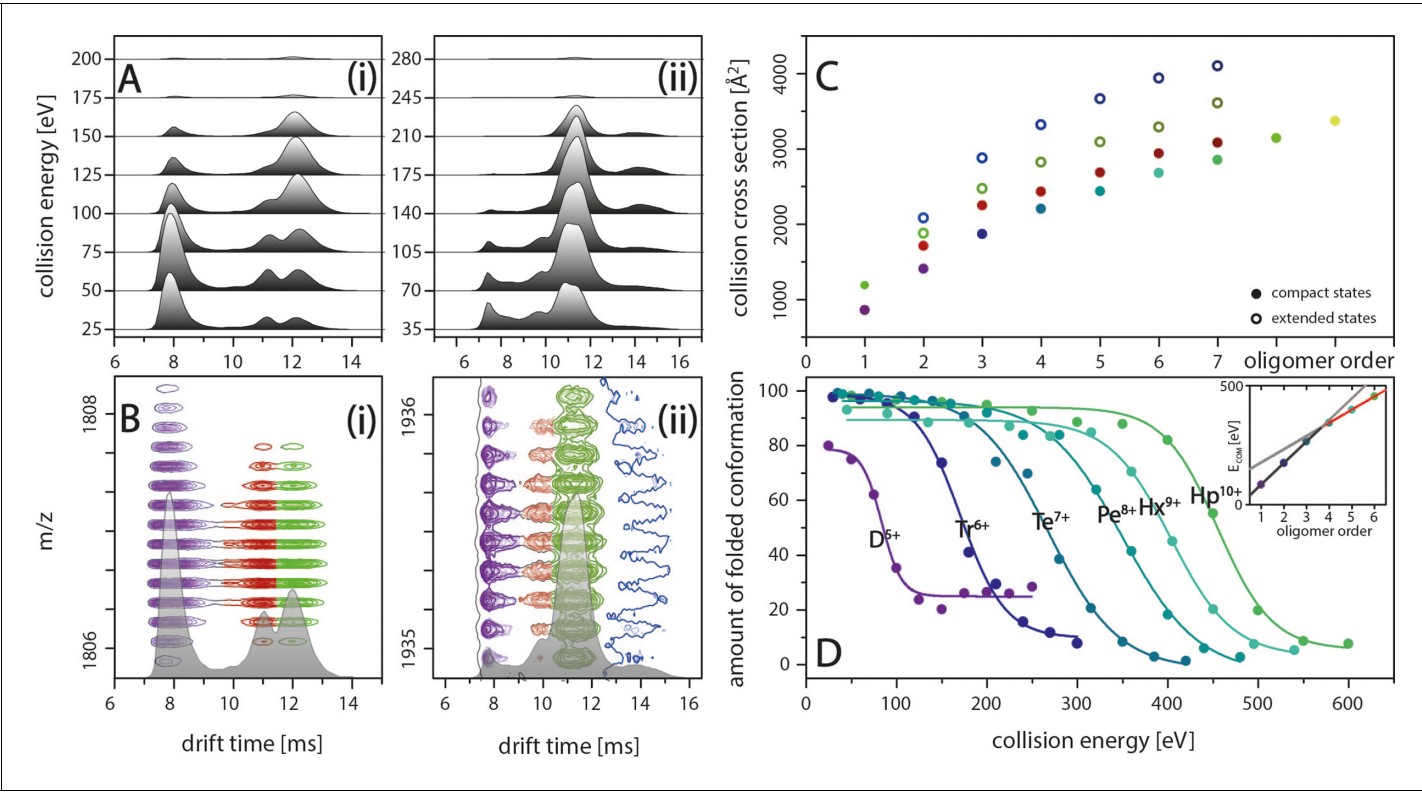

**Figure 2.** IM spectra of the five-times charged Aβ42 dimer (i) and of the seven-times charged trimer (ii) for different CE. The structured MB conformation at a drift time of 8 ms shows an unfolding product (drift time above 12 ms), whereas the structured DB conformation (drift time of 11 ms) does not unfold before undergoing collision induced dissociation. (B) Driftscope plots of the five-times charged dimer (i) and the seven-times charged trimer (ii), shown in A. The colors indicate the two dimer structures (purple and red) as well as the unfolding product (green). For the trimer, a second unfolding product is detected (dark blue). (C) Increased lab frame energies can lead to unfolding of both structured conformations for each oligomer: CCS for the four species (two compact and two extended states) for each of the different oligomers, averaged over all charge states. Color code matches with B+D. (D) CIU of the different MB oligomers depending on CE, corrected for the charge states of the oligomers. The stability of the oligomers increases with their size at different rates below and above the tetramer (inset).

derived CCS. In both cases, we used the respective dimer CCS as a spatially isotropic peptide as reference for the isotropic growth model. We assess the growth behavior separately for both data sets (depicted in red or blue in *Figure 1B*, respectively). The data set with the lower CCS values (red in *Figure 1B*) clearly corresponds to the isotropic growth model for the lower oligomers, while unambiguous assignment to either model is not possible for higher oligomers. The experimental data points for each oligomer show a less homogeneous distribution of CCS than the blue dataset, as seen by the larger error bars, suggesting less clearly defined structures. Assessment of the growth behavior of the data set with the higher CCS values (indicated in blue in *Figure 1B*) shows that the isotropic growth curves describes our data up to the tetramers well, after which the experimental data clearly deviates from the isotropic growth model predictions. The CCS of these higher order oligomers have a different dependence on the oligomeric Aβ42 state. CCS for oligomers larger than a tetramer can be perfectly fit by a linear growth model. Thus, our data show a change in the arrangement of oligomers from a more globular to a linear growth with oligomers larger than tetramers following the linear growth model. This suggests that less than four peptides do not stabilize each other sufficiently to form a stable structure. This finding is especially interesting in the light of predictions based on cryo-EM and solid-state NMR studies, which proposed a minimum of five Aβ42 molecules required to form a regular fibril structure (*Gremer et al., 2017*). Our observation of linear growth from tetramer onward is encouraging for the aim to correlate our findings with structures determined by solid-state NMR (*Colvin et al., 2016*; *Wälti et al., 2016*) or by cryo-EM (*Gremer et al., 2017*), bearing in mind that structural polymorphism has been reported both with regard to the microscopic structure and morphology of fibrils. The common feature of the reported

structures is a fibrillary structure with S-shaped $A\beta_{42}$ forming a dimer as base for the fibril growth. These imply that $A\beta_{42}$ peptides have two different ways of interacting with each other – a planar interaction, forming a planar dimer or a stacking interaction, allowing growth along the fibril growth axis. The two different CCS we observe for each oligomer also suggest more than a single growth mechanism (*Bleiholder et al., 2011*). Both sets of CCS show a very similar increase of CCS per oligomer (compare the linear fits in *Figure 1B*), suggesting growth via the same S-shaped $A\beta_{42}$ monomers. But the different y-interceptions indicate a different growth base leading either to a single-base oligomer or an oligomer building on a dimeric base, known as β-sheet zippers (*Bleiholder et al., 2011*). Thus, we assign the oligomers giving the larger set of the experimentally determined CCS to $A\beta_{42}$ peptides in DB arranged structures. For the second set of CCS, the overall values are smaller than could be explained by structures with a DB $A\beta_{42}$ - they show less homogeneous structures and therewith less clearly follow a linear growth model. This corroborates the assignment of these oligomers to MB arranged structures, which could be less well defined than the DB arranged counterparts. In summary, we assign the two different CCS to arise from either growth via singular $A\beta_{42}$ peptides starting from an $A\beta_{42}$ monomer or alternatively stacking onto a planar $A\beta_{42}$ dimer (*Figure 1C*). Based on these findings, we conclude that two oligomerization pathways exist, one via DB which are on pathway to the stable fibrillary structures as suggested by NMR and cryo-EM measurements (*Colvin et al., 2016*; *Gremer et al., 2017*; *Wälti et al., 2016*). We suggest that the other pathway, where oligomers are formed via MB, can potentially lead to amorphous aggregates but not stable fibrils. This model of two growth pathways is interesting in the light of studies showing that a single mutation of $A\beta_{42}$ can change the aggregation pathway leading to amorphous aggregates only, rather than fibrils, as was shown for the $A\beta_{42}$ mutant F19P (*de Groot et al., 2006*). If our model is correct and the DB conformation is essential for the formation of fibrils, we would expect it to be missing for this mutant. In supplementary data *Appendix 1—figure 3A*, we show the differences in the IM spectrum of F19P as opposed to wild-type $A\beta_{42}$, for the example of $(A\beta_{42})_2^{5+}$ ($D^{5+}$). In support of our model, we can assign the structured and the extended conformation of the MB conformation, but no signal that corresponds to a DB arrangement. The respective TEM images shows amorphous aggregation (*Appendix 1—figure 3B*). These findings show that we can determine with native MS the structural development of small $A\beta_{42}$ oligomers to occur via two pathways, MB and DB stacking.

The observed growth pattern changes for DB oligomers larger than a tetramer, which represents a landmark in $A\beta_{42}$ oligomer growth, at which the $A\beta_{42}$ peptides support each other enough to build structures which follow linear growth.

## Oligomer stability is size-dependent

The IM spectra in *Figure 2A* show the two structured species of $D^{5+}$ at drift times around 8 ms and 11 ms, representing the DB and MB arrangements, respectively. At higher CEs, the IM spectra additionally reveal unfolding for the $A\beta_{42}$ oligomers whereby, despite peak overlap, the oligomeric state can be unambiguously assigned due to isotopic resolution of the corresponding driftscope plots (*Figure 2B*). To further investigate this effect, we submitted the $A\beta_{42}$ oligomers to increasing collisional activation in the collision cell, leading to CIU of the peptide ions. This unfolding process reports on the intramolecular stability within the polypeptide chain and can vary if these interactions are changed by ligand binding, mutations or changes in pH (*Dixit et al., 2018*). Only one species of the two $A\beta_{42}$ dimer conformations of $D^{5+}$ shows unfolding upon collision with inert gas molecules. A signal correlating to the unfolding product of the DB conformation is missing for this charge state. Thus, the CE applied is sufficient to unfold the MB but not the DB conformation before collision-induced dissociation (CID) of the dimer at 125 eV (*Figure 2A(i)*). (For signal intensities in dependence of different CE see as well Figure 4D(i)). For all oligomers, the higher charge states show unfolding for both species (*Figure 2A(ii)* as example for $Tr^{7+}$). *Figure 2C* shows the CCS for the two observed structured complexes as well as two unfolding products for all oligomers. Supplementary data of *Appendix 1—figure 4* depicts this in more detail for the example of the nine-times charged pentamer. The intermolecular stability shifts with oligomeric size. Observing these shifts within the MB conformations allows us to monitor the stability increase of oligomer growth along the growth axis. For the dimer, 50% unfolding is reached at 85 eV (*Figure 2D*). The CE needed for unfolding increases for every oligomer with the heptamer unfolding at 450 eV. Interestingly, the energy gaps between the CE sufficient to unfold the different oligomers are equidistant from dimer to tetramer

and again from pentamer, but with a decreased energy gap. In line with this, the red line in the inset in *Figure 2D* shows equidistance of the energy gaps for oligomers larger than a tetramer. Compared to this trend, which reflects the stability against unfolding for the larger oligomers, the CEs needed to unfold oligomers smaller than a tetramer (gray line in inset of *Figure 2D*) are surprisingly low. This supports our earlier notion that an ordered stable fibrillary structure evolves only after the tetramer, in line with previously proposed theoretical models (*Gremer et al., 2017*).

## MB and DB structures

After establishing that $A\beta_{42}$ can oligomerize via two pathways with an MB or DB base, we attempted to shed more light on the resulting structures. We made use of protein database (PDB) entries of $A\beta_{42}$ fibrils to calculate theoretical CCS for the different oligomers. To distinguish between fibrillary growth via stacking based on an $A\beta_{42}$ monomer and a growth structure with a planar arranged DB $A\beta_{42}$, we dissected the PDB structure and constructed two sets of $A\beta_{42}$ oligomers. Each structure was subjected to MD simulation for up to 1 µs to allow rearrangement of the fibril fragments, taking into account the effects of the charges carried due to the ESI process. The resulting structures support our earlier notion regarding development from more isotropic structures toward more stable fibril-like structures for the higher oligomeric states (example structures shown in supplementary data *Appendix 1—figure 5* and *6*). We analyzed the obtained structures for different structural elements, such as coiled or β-sheet structure (see supplementary data *Appendix 1—figure 6*). The most striking feature is the increase of the percentage of β-sheet motive, the bigger the oligomers. This is the case for all tested PDB structures as well as both types of oligomers: the MB and the DB based structures. The higher β-sheet content correspond to the higher stability, seen for the higher oligomers. These findings correlate well with insights from infra-red spectroscopic analysis of IMS-MS separated gas phase ions of smaller amyloidic peptides. As well an increase in β-sheet content was revealed for higher oligomers, showing that stable structural features remain in the gas phase (*Seo et al., 2017*). Using the MD structures, theoretical CCS were calculated (see supplementary information for detailed description). While it should be noted that we are comparing gas phase structures here, which stem from different solution structures, the found correlations are noteworthy. The best correlation was found with CCS obtained from PDB structure 5OQV (see supplementary data *Appendix 1—figure 7*). The theoretical CCS correlate well with the experimentally determined values. They show the same increase of CCS per additional $A\beta_{42}$ monomer for both sets of values, supporting the S-structured $A\beta_{42}$ as the building block in each case. Additionally, the theoretical CCS for the fibrillary DB oligomers correspond very well to the overall values of the experimentally determined CCS.

## CLR01 inhibits $A\beta_{42}$ oligomerization

Our MS investigations of the different aggregation pathways of $A\beta_{42}$ oligomers might provide a rationale for aggregation inhibition by low-molecular-weight inhibitors. We therefore investigated aggregation inhibition with respect to the inhibitor CLR01 (molecule depicted in *Figure 4A (ii)* and supplementary data *Appendix 1—figure 8*). As ESI only allows detection of the first few oligomers and does not report on time-dependent changes, we performed time-resolved laser-induced liquid bead ion desorption mass spectrometry (LILBID-MS) measurements. For this purpose, aliquots of $A\beta_{42}$ were incubated at 22°C in the presence and absence of inhibitors. *Figure 3* and supplementary data *Appendix 1—figure 9* show the development of the aggregation process of $A\beta_{42}$ without and with a 4-fold excess of CLR01. For $A\beta_{42}$ alone, aggregation development suggests a constant progression from monomers up to dodecamers after 200 min. In comparison, the highest oligomeric state present 200 min after addition of CLR01 is a hexamer. The kinetics of $A\beta_{42}$ aggregation, show the inhibitory effect of CLR01 (*Figure 3*). The inhibiting effect is comparable to that observed for OR2 (*Stark et al., 2017*; *Appendix 1—figure 8*).

## Inhibitors stop origin of dimer-based oligomers

We further measured changes in IM of $A\beta_{42}$ in the presence of fourfold excess of CLR01 and OR2 (supplementary data *Appendix 1—figure 8*). For both ligands, we detect IM peaks for the MB $A\beta_{42}$ dimer binding to a single ligand molecule. This peak appears at higher drift times compared to the $A\beta_{42}$ dimer. In sharp contrast, no signal can be observed for the DB $A\beta_{42}$ dimer, showing the

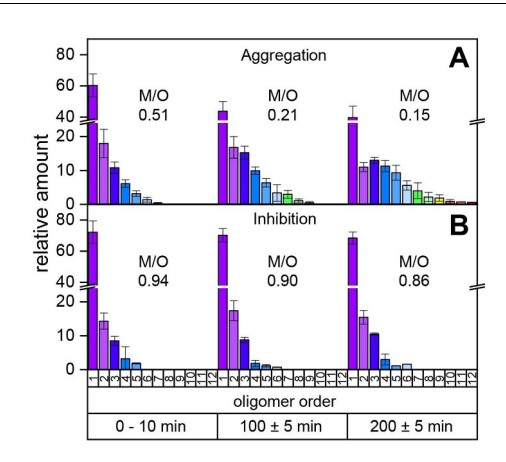

**Figure 3.** Time-resolved measurements of the oligomerization of Aβ$_{42}$ incubated at a temperature of 22°C. The aggregation of the monomerized Aβ$_{42}$ sample was tracked for 200 min. The signals of the corresponding oligomers were averaged for the period of 10 min. The observable oligomers are indicated in the color code according to *Figure 2D*. LILBID-MS measurements of the full time period (supplementary data *Appendix 1—figure 9*) show the time-dependent development of the first Aβ$_{42}$ oligomers (A), which is hindered in the presence of CLR01 (B). The relative intensity of the distinct oligomers are shown for different time points, normalized to the spectrum's total intensity. The monomer over oligomer ratio (M/O) values reflect the development of the Aβ$_{42}$ oligomerization.

suppression of this conformer by both inhibitors. (*Figure 4A(i) and (ii)* and *Appendix 1—figure 9A*). Both inhibitors having the same effect (inhibition of DB dimer formation) is especially interesting in view of the differences of these inhibitors. The inhibitor OR2 is designed to interfere with the central hydrophobic KLVFF region in Aβ$_{42}$ (*Austen et al., 2008*), while CLR01 was shown in the monomeric unfolded state of Aβ$_{42}$ to interact in the area of R5, K16, and K28 (supplementary data *Appendix 1—figure 10*; *Zheng et al., 2015*; *Sinha et al., 2011*; *Schrader et al., 2016*; *Sinha et al., 2012a*; *Sinha et al., 2012b*), suggesting a different mechanism of action. The disappearance of one of the basic modules for Aβ$_{42}$ oligomerization (the DB module) upon binding of ligands shows the ligands' influence on the quaternary structure of Aβ$_{42}$, which must be achieved via different changes in tertiary structure: Such changes in tertiary structure can go along with a change in a protein's stability, which can be observed as a different reaction toward collisional activation. *Figure 4* depicts the effect of increasing CE on free and CLR01-bound D$^{5+}$. (The respective experiments with OR2 are shown in supplementary data *Appendix 1—figure 12*). Both species, the bound and the unbound one, dissociate into Aβ$_{42}$ monomers (M$^{2+}$ and M$^{3+}$) (*Figure 4B(i) and (ii)*). CLR01 can remain bound to one Aβ$_{42}$ monomer. To look at this in more detail, CIU heat maps (*Figure 4C*) of free and CLR01 bound D$^{5+}$ and the respective intensity plots (*Figure 4D*) show the effect that increasing CE has on the dimer. An enhanced CE leads to CIU, as seen by an increase of drift time in *Figure 4C*, and CID observable by the disappearance of the dimer signal. For easy comparison significant CEs (leading to 50% CIU or CID) are summarized in the graphs in *Figure 4E(i) and (ii)*. For the different species, these CEs deviate noticeably, revealing differences regarding the dimers' stabilities toward unfolding or dissociation (*Figure 4(i) and (ii)* as well as supplementary data *Appendix 1—figure 11*), which will be explained in detail in the following: For unbound Aβ$_{42}$, we observe differences between the MB and the DB structures. The increase of CE to 85 eV causes 50% unfolding of the MB dimer (*Figure 4D(i)* and supplementary data *Appendix 1—figure 11B*), and higher energies are needed for dissociation. In contrast, the same 85 eV is already sufficient for 50% dissociation of the DB dimer, which interestingly dissociates without any prior unfolding (*Figure 4C(i) and D(i)*). For ligand bound Aβ$_{42}$, we can only observe the effects of CE for the MB dimers, as both ligands prevent the formation of the DB dimer. Both ligands increase the CE needed for 50% CID slightly (for CLR01 shown in *Figure 4B(i)* and supplementary data *Appendix 1—figure 11C*). The unfolding process is not hampered upon OR2 binding (a similar amount of CE leads to 50% CIU with or without OR2 bound) while CLR01 interaction stabilizes the intramolecular structure, so 50% CIU cannot be achieved as maximally 15% of the proteins unfold prior to complete D$^{5+}$ dissociation (*Figure 4C(ii) and D(ii)*). While OR2 does not alter the unfolding tendencies of Aβ$_{42}$, CLR01 strongly stabilizes intramolecular interactions within an Aβ$_{42}$ peptide, while leaving intermolecular interactions in terms of aggregation axis growth unaltered. These results show that both ligands have a different mechanism by which they hinder the planar DB interaction and therewith suppress DB formation, leading to the formation of amorphous aggregates instead of fibrils, as seen in TEM images (*Figure 5*). This is especially interesting, as it shows that drugs, which inhibit the fibril

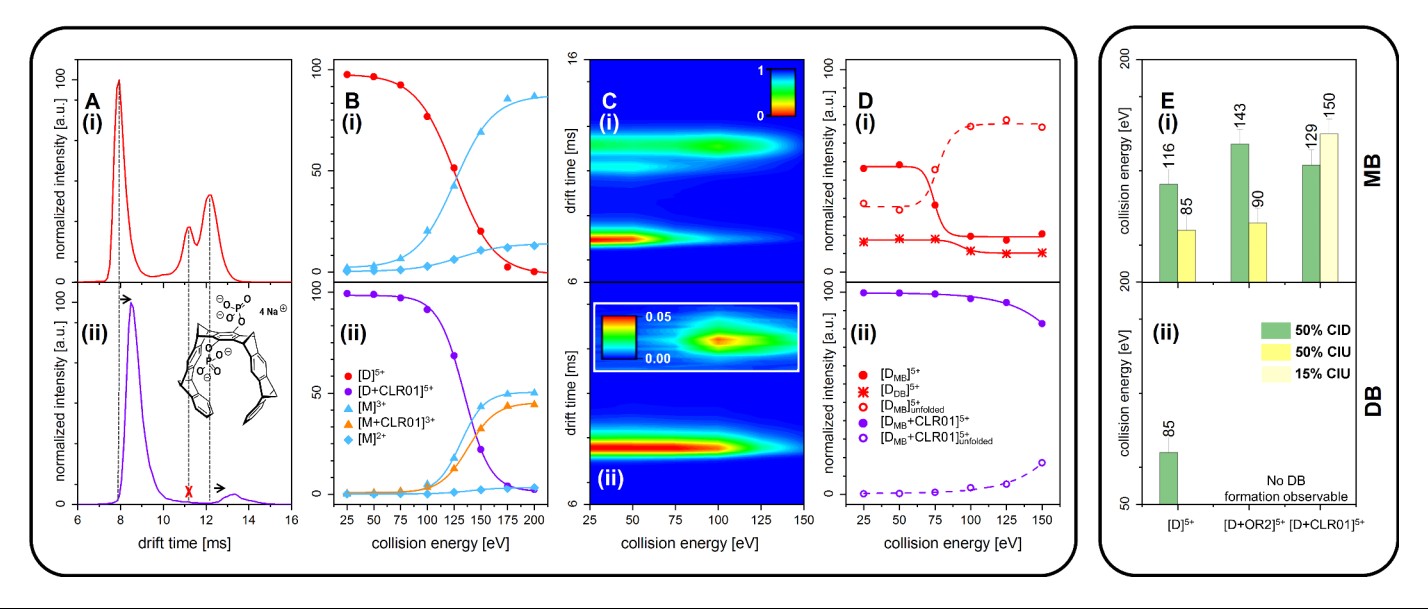

**Figure 4.** CID and CIU experiments comparing free Aβ42 (top row (i)) and CLR01-bound Aβ42 (bottom row (ii)). (**A**) (i) and (ii) show a comparison of the IM spectra of the 5-times charged dimer of pure and CLR01-bound Aβ42, respectively. (**B**) (i) and (ii) show CID experiments of the D5+ of free and CLR01-bound Aβ42, respectively. (**C**) (i) and (ii) show a heat map of CIU experiments in dependence of CE increase for D5+ of free and CLR01-bound Aβ42, respectively. In (**C**) (ii) the signal for the dependence of the drift time on CE for the extended unfolding product of the MB dimer is scaled up by a factor of 20. (**D**) (i) and (ii) show an intensity plot of the peaks visible in the CIU experiment of C. (**E**) CE which lead to 50% CIU or CID for D5+ with or without one bound ligand for the MB (i) and DB (ii) conformation. In case of CLR01, the maximally observed amount of CIU product is 15% due to prior CID. Errors given are a conservative estimate of three repeats. Data shown in supplementary data *Appendix 1—figure 11*.

formation of Aβ42 might interact via different mechanisms, but can still be assessed by simply monitoring their ability to suppress the DB dimer.

## Discussion

Our data support the formation of two arrangements for the first step in fibril formation of Aβ42, the major cause for progression of the AD. We can characterize stability and formation kinetics of the two fundamentally important Aβ42 dimers which form the basis for Aβ42 oligomers. According to previously characterized fibril structures, (*Colvin et al., 2016*; *Gremer et al., 2017*; *Wälti et al., 2016*; *Xiao et al., 2015*) dimers can either form C2 symmetric Aβ42 structures (DB form) or exhibit only translational symmetry along the aggregation axis (MB). These dimers serve as base for further aggregation. Our IM data reveals that only oligomers larger than a tetramer support each other sufficiently to adopt a stable structure. The structures of smaller oligomers are less defined, tending to adjust to a more globular overall shape. Nevertheless, well-defined inter- and intramolecular interactions are already formed, as seen by CIU and CID data. This is relevant, as it shows that the foundation for the fibril formation is already laid in the first dimer. Inhibitors or mutations that influence the formation of the aggregation base will lead to different oligomerization processes. *Figure 5* summarizes our findings for Aβ42. Despite different binding sites of the two different inhibitors (ionic/hydrophobic), it is remarkable that both disturb the formation of the S-shaped Aβ42 structure enough to hinder the formation of the planar DB Aβ42, which is a pre-requisite for the formation of stable fibrils. Without this DB dimer the remaining Aβ42 monomers stack axially in a less stable manner. This leads to the inhibition of the on-pathway fibrillary growth as we could show with time resolved measurements. We conclude that an undisturbed S-shaped structure of the Aβ42 monomer is relevant for an orderly evolvement of large oligomers and fibrils.

TEM images show an off pathway amorphous aggregation instead of an on-pathway fibrillary one in all the cases where we found the DB arrangement to be missing (Aβ42 with OR2 or CLR01, as well as for the Aβ42 mutant F19P). Those amorphous aggregates are known to be non-toxic (*Lee et al.,*

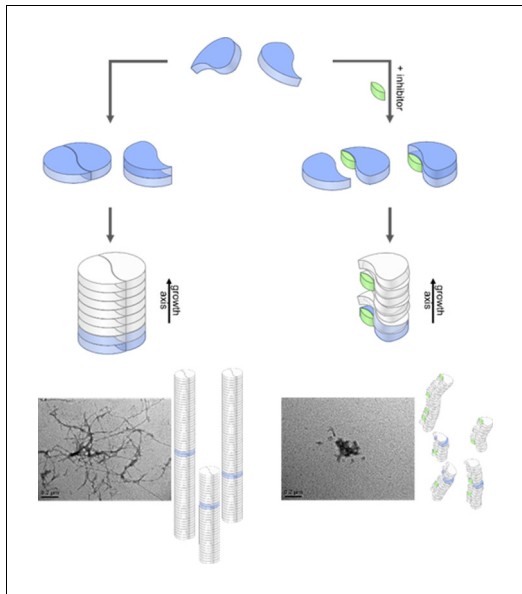

**Figure 5.** Aggregation pathways and inhibition by the low molecular weight inhibitor CLR01. Monomeric Aβ$_{42}$ can oligomerize via two separate ways, either by an axial stacking of monomers which results in amorphous aggregates or by DB stacking which results in fibrils. The pathway involving stacking of dimers is inhibited by CLR01. Inserts show TEM images for free (left image) and CLR01-influenced (right image) Aβ$_{42}$.

*2017*) which explains the detoxifying effect of CLR01 (*Sinha et al., 2012a*). In summary, we could show by combining our MS and IMS results that the DB conformation is the seed for the toxic on-pathway Aβ$_{42}$ oligomers. This aggregation module is suppressed by the inhibitors used herein. This makes the suppression of the DB Aβ$_{42}$ dimer conformation a prime target for future drug development.

## Materials and methods

### Sample preparation

Beta-amyloid: For the mass spectrometry (MS) experiments both recombinant and synthetic Aβ$_{42}$ was analyzed. Cloning, Expression, and Purification of recombinant Aβ$_{42}$: The Aβ$_{42}$ DNA sequence, codon-optimized for *E. coli*, used to construct the synthetic gene was the same as that described by *Weber et al., 2014*, which was based on the previously published construct by *Walsh et al., 2009* The gene was constructed with BSaI and XbaI restriction sites and cloned into a pE-SUMOpro expression vector which yielded a construct consisting of a His-6 tagged N-terminal SUMO fusion protein attached to the N-terminus of Aβ$_{42}$. SUMO-Aβ$_{42}$ plasmid DNA was transformed into competent *E. coli* DH5α subcloning cells (Invitrogen, Carlsbad, CA, USA) by heat shock, amplified and extracted via miniprep according to the manufacturer's guidelines (Qiagen, Hilden, Germany). Expression of uniformly labeled $^{15}$N- or $^{13}$C$^{15}$N-SUMO-Aβ$_{42}$ involved selecting a single colony of freshly transformed *E. coli* BL21 DE3 expression cells to inoculate 5 mL LB-medium with ampicillin, from which 100 μL was used to inoculate 100 mL LB-medium with ampicillin. The cells from the 100 mL overnight starter culture were harvested by centrifugation (6000 g, 10 min, 25°C), suspended in M9 minimal medium and used to inoculate 1 L M9 minimal medium enriched with $^{13}$C-glucose and/or ($^{15}$NH$_4$)$_2$SO$_4$ (1 g/L). The cultures were grown at 37°C until an OD$_{600}$ of 0.6 was reached, whereupon expression of SUMO-Aβ$_{42}$ was induced with 1 mM isopropyl β-D-1-thiogalacto-pyranoside. The cells were harvested by centrifugation (4000 g, 30 min, 4°C) after 5 hr of incubation at 37°C (5.5 g/L wet cell mass) and stored at −80°C. Purification of $^{15}$N- or $^{13}$C$^{15}$N-Aβ$_{42}$ was performed in part as previously described with some modifications (*Weber et al., 2014*; *Walsh et al., 2009*). The cell pellet was resuspended in IB buffer - 10 mM Tris, 175 mM NaCl, 1 mM DTT, 1 mM EDTA, pH 8.0, sonicated (5 × 30 s at 50% power) on ice and subsequently centrifuged (6000 g, 10 min, 4°C). The supernatant was decanted, the pellet resuspended in IB buffer and the sonication and centrifugation procedure was repeated a further time. The pellet was resuspended and washed (IB buffer) using a dounce homogenizer, then centrifuged (6000 g, 10 min, 4°C). This procedure was repeated three times. The (white IB) pellet was solubilized and sonicated (3 × 30 s at 50% power) in lysis buffer (either 6 M GdnCl, 100 mM NaPi, 20 mM Tris, pH 8.0 or 8 M Urea, 100 mM NaPi, 10 mM Tris, pH 8.0), then centrifuged (6000 g, 10 min, 4°C). The supernatant was diluted to a final urea or GdnHCl concentration of 0.5 M, after which SUMO protease Ubiquitin like protein-1 (Ulp1) was directly added and allowed to cleave over 48 hr at 4°C or 2 hr at room temperature. Cleavage of SUMO-Aβ$_{42}$ was verified by SDS-PAGE. The cleavage mixture was EtOH precipitated and the pellet was dissolved in 70% formic acid, passed through a 0.45 μm PTFE filter and subjected to HPLC chromatography (Perfectsil RP4 column, 4.6 × 250 mm, 300 Å C4 5 μM, 60°C, 1.3 mL/min; 230 nm fixed wavelength detection). A 30–80% gradient was run where the solvent was 400 mM HFIP/TEA, pH 7.6 and the eluent MeOH. Fractions were collected on ice, analyzed by SDS-PAGE

and MALDI and the $A\beta_{42}$-containing fractions dried at 4°C in a concentrator plus (Eppendorf, Hamburg, Germany) to produce a peptide film, which was stored at −80°C. The final yield of $A\beta_{42}$ after HPLC purification was approximately 3.5 mg/L. Expression and purification of the SUMO-protease Ulp1: Plasmid DNA encoding the residues 403–621 of *Saccharomyces cerevisiae* Ulp1 with a His-tagged N-terminal was used. The pET-28b plasmid encoding Ulp1 was transformed into *E. coli* DH5α subcloning cells for the amplification of the plasmid DNA, and into BL21 DE3 cells for expression. For expression, 1 L LB medium selective for ampicillin (100 µg/mL ampicillin) was inoculated with cells to an $OD_{600}$ of 0.1 and grown at 37°C and 180 rpm. Expression was induced at an $OD_{600}$ of 0.6 by the addition of 1 mM IPTG and growth was continued for a further 4–5 hr. The cells were harvested by centrifugation (4000 g, 20 min, 4°C), frozen and stored at −80°C (wet cell mass 5.2 g/L). Ulp1 SUMO-protease cell pellets were suspended in native buffer (50 mM NaPi, 300 mM NaCl, 10 mM Imidazole, 10 mM β-ME, pH 8.0) and subjected to sonication (15 cycles, 50% power) to lyse the cells. The lysate was centrifuged and the clear supernatant loaded to a pre-equilibrated 5 mL Ni-NTA HisTrap column (Qiagen) whereby an elution gradient with increasing imidazole concentration resulted in the elution of the Ulp1 at 300 mM Imidazole. A yield of 65 mg/L culture was determined using the theoretical extinction coefficient of 28590 $M^{-1}$ $cm^{-1}$ at 280 nm. Synthetic $A\beta_{42}$ was prepared as published before (*Stark et al., 2017*). Briefly, lyophilized $A\beta_{42}$ was purchased from AnaSpec, USA (#24224). The peptide was solvated in HFIP containing 3% concentrated $NH_3$. After incubation for 5 min, the solution was aliquoted in protein low-bind Eppendorf tubes and the solvent was evaporated using a SpeedVac RVC 2–18 (Christ, Osterode, Germany) concentration system. The remaining peptide film was stored at −80°C. For usage, the peptide film was solvated using DMSO (1%) and 50 mM $NH_4OAc$ buffer at pH 7.4 (if not stated otherwise) to a final peptide concentration of 50 µM. It was stored on ice until beginning the kinetic measurements. Incubation of the $A\beta_{42}$ sample was done at room temperature of 22°C in its respective Eppendorf tube. Both $A\beta_{42}$ species, recombinant and synthetic, behave identically regarding their structure as detected by ESI ion mobility spectrometry (IMS) as well as in oligomerization detected by LILBID-MS (supplementary data *Appendix 1—figure 13*). CLR01 was synthesized and then solvated at a concentration of 2 mM in deionized water (*Fokkens et al., 2005*). For testing the influence of CLR01 on $A\beta_{42}$, the molecule was added in fourfold excess to the peptide. OR2 was synthesized as published before via solid-phase synthesis (*Cernescu et al., 2012*; *Matharu et al., 2010*). For storage, the molecule was solvated at a concentration of 20 mM in DMSO which was diluted to fourfold excess regarding $A\beta_{42}$ for MS analysis.

## Mass spectrometry analysis

Electrospray ionization mass spectrometry (ESI-MS) was performed on a Synapt G2S (Waters Corpn., Wilmslow, Manchester, UK) equipped with a high-mass quadrupole upgrade. Pd/Pt sputtered nESI tips were pulled in house from borosilicate glass capillaries on a Flaming/Brown Micropipette Puller (P-1000; Sutter Instrument Co.). $A\beta_{42}$ was analyzed in positive ion mode using a capillary voltage of 1.9 kV. The rest of the settings for MS analysis were adjusted as following: cone voltage 100 V at an offset of 80 V, 20°C source temperature. The instrument was calibrated by a conventional CsI solution. All experiments were as well performed in negative ion mode. The observed spectra support our conclusions (*Appendix 1—figure 2*) but show lower signal intensity/quality. Therefore, in this publication we present the data achieved in positive ion mode. IM experiments were done on the same instrument using a traveling wave setup with a wave height of 40 V, a travelling wave velocity of 700 m/s, a nitrogen gas flow of 90 mL/min, drift cell pressure of 3.5 mbar. To calculate CCS values, the instrument was calibrated using cytochrome c, apo-myoglobin and ubiquitin under denaturing conditions (*Appendix 1—figure 14*). Collision-induced unfolding (CIU) and -dissociation (CID) experiments were done by ramping the trap collision energy (CE) in steps of 5 V from 5 to 50 V. MS-MS was performed to detect the appearance of the dissociation products of the CID experiment. Thereby, the five-times charged dimer peak, which is discussed in the manuscript, was selected at m/z = 1806 using an LM resolution of 12 and a HM resolution of 15. Laser-induced liquid bead ion desorption MS (LILBID-MS) measurements were performed as previously published (*Stark et al., 2017*; *Cernescu et al., 2012*). Thereby, for each time-point 4 µL of the $A\beta_{42}$ solution described above was injected into the droplet generator (MD-K130 from microdrop Technologies GmbH, Norderstedt, Germany) separately. The ions produced by the LILBID process were analyzed as negative ions by time-of-flight. Four spectra were recorded at laser intensities below the damage threshold of

the $A\beta_{42}$ oligomers. Each of those spectra is an average of the analysis of 500 droplets. Mass-calibration was achieved by recording spectra of bovine serum albumin. The oligomeric state of the $A\beta_{42}$ sample was determined in terms of the monomer-to-oligomer (M/O) ratio by calculating the ratio of the intensities of the $A\beta_{42}$ peaks:

$$M/O = \frac{I_1}{\sum n \cdot I_n} \tag{1}$$

The values of those four spectra were averaged to obtain a time depending M/O value. Data analysis of ESI-MS and IMS experiments was done using the software MassLynx V4.1 and UniDec (*Marty et al., 2015*). CCS calibration was performed according to the protocol by *Ruotolo et al., 2008*. To process the LILBID-MS spectra, the software Massign was used (*Morgner and Robinson, 2012*). Using this software, the raw spectra were calibrated, smoothed and background subtracted. Visualization of the results was performed using (Origin 2018 OriginLab Corporation, USA).

## MD simulations

To correlate the two sets of experimental CCS values we obtained for the different $A\beta_{42}$ oligomers with potential structures, we derived structures for the different oligomers from existing $A\beta_{42}$ PDB entries. We selected the PDB entries 5OQV and 2NAO which represent $A\beta_{42}$ fibrillar structures with a dimer base, deduced from cryo-EM and solid-state NMR experiments, respectively (*Gremer et al., 2017*; *Wälti et al., 2016*). Cropping these allowed us to obtain two sets of structures (dimer based (DB) as well as monomer based (MB)) of the respective small oligomers observed in the ESI-MS experiments. The fibril PDB structure 5OQV is a 9-mer with dimeric base, permitting to extract oligomers until 9-mer for the DB structures and oligomers until 5-mer for the MB structures. 2NAO is a 6-mer, accordingly allowing for DB 6-mer and MB 3-mer. Experience has shown that CCS values computed straight from X-Ray, NMR or cryo-EM structures are usually not identical to experimental CCS as received from ESI gas phase ions (*Heo et al., 2018*). We took potential alterations of the protein structures in the gas phase such as charge driven distortion or compaction due to self-solvation during the ESI process into account by performing vacuum MD simulation using Gromacs 5.0.7 (*Berendsen et al., 1995*; *Pronk et al., 2013*; *Van Der Spoel et al., 2005*). Preceding equilibration of the cropped PDB structures in water were performed based on Pujol-Pina et al., but with the simulation time doubled to 10 ns (*Pujol-Pina et al., 2015*). Prior to vacuum MD simulations charge effects were taken into account by placing charges onto the isolated structures, according to the most abundant experimental gas phase charge state. Individual charge states were adjusted by manually protonating basic residues according to the experimentally observed species. For each oligomer/charge state combination, a set of structures was generated with the respective number of protons randomly distributed to different basic residues to reflect an ensemble of structures. These structures were then submitted to computation for simulation for 1 µs or 200 ns, respectively, as detailed in supplementary data *Appendix 1—figure 1B*, using the AMBER99SB-ILDN force field. The resulting structures were used to perform CCS calculations using the ImoS software (*Larriba and Hogan, 2013*). Nitrogen was selected as a buffer gas, and pressure as well as temperature were set according to experimental conditions. The DSSP algorithm was used to assign the content of secondary structure of the resulting structures (*Kabsch and Sander, 1983*).

## TEM measurements

50 µM $A\beta_{42}$ were incubated for 48 hr in 50 mM $NH_4OAc$, 1% DMSO at pH 7.4. Samples were spotted on carbon-coated copper grids and negative stained with 2% uranyl acetate. Pictures were recorded with a Philips CM 12 with a magnification of 66000.

## Acknowledgements

The authors thank M Vabulas, J Wachtveitl and C Robinson for critical reading of the manuscript. We thank M Göbel for support on purification of synthetic beta-amyloid, J Hildenbrand and C Hamerla for advice on MD calculations. We thank Marion Basoglu for supporting to record TEM images. We gratefully acknowledge funding for TL and JM by DFG via GRK1986 and from the State of Hesse in the LOEWE Schwerpunkt GLUE for RZ. NM was supported by Cluster of Excellence Frankfurt

(Macromolecular Complexes) and received funding from the DFG via a Heisenberg professorship. TS gratefully acknowledges generous financial support of the CRC1093 'Supramolecular Chemistry on Proteins' by the DFG (Deutsche Forschungsgemeinschaft). Work at BMRZ (Centre for Biomolecular Magnetic Resonance at Goethe University) is supported by the HMWK (Hessian Ministry for Science and the Arts). This project has received funding from the European Union's Horizon 2020 research and innovation programme under grant agreement No 871037 (iNEXT-discovery).

## Additional information

### Funding

| Funder | Grant reference number | Author |
|---|---|---|
| State of Hesse | LOEWE Schwerpunkt GLUE | Rene Zangl |
| Cluster of Excellence Frankfurt | Macromolecular Complexes | Nina Morgner |
| Deutsche Forschungsgemeinschaft | Heisenberg professorship | Nina Morgner |
| Deutsche Forschungsgemeinschaft | GRK1986 | Tobias Lieblein Janosch Martin |
| Deutsche Forschungsgemeinschaft | CRC1093 'Supramolecular Chemistry on Proteins' | Thomas Schrader |
| Horizon 2020 - Research and Innovation Framework Programme | No 871037 (iNEXT-discovery) | Harald Schwalbe |
| State of Hesse | BMRZ | Harald Schwalbe |

The funders had no role in study design, data collection and interpretation, or the decision to submit the work for publication.

### Author contributions

Tobias Lieblein, Conceptualization, Data curation, Formal analysis, Supervision, Investigation, Methodology, Writing - original draft, Project administration, Writing - review and editing; Rene Zangl, Data curation, Software, Formal analysis, Validation, Investigation, Visualization, Methodology, Project administration, Writing - review and editing; Janosch Martin, Formal analysis, Investigation; Jan Hoffmann, Software, Formal analysis, Investigation; Marie J Hutchison, Resources, Investigation, Writing - review and editing; Tina Stark, Elke Stirnal, Resources; Thomas Schrader, Resources, Writing - review and editing; Harald Schwalbe, Supervision, Funding acquisition, Writing - review and editing; Nina Morgner, Conceptualization, Supervision, Funding acquisition, Methodology, Writing - original draft, Project administration, Writing - review and editing

### Author ORCIDs

Tobias Lieblein http://orcid.org/0000-0002-6497-1733
Jan Hoffmann https://orcid.org/0000-0002-0770-9886
Thomas Schrader https://orcid.org/0000-0002-7003-6362
Harald Schwalbe https://orcid.org/0000-0001-5693-7909
Nina Morgner https://orcid.org/0000-0002-1872-490X

### Decision letter and Author response

Decision letter https://doi.org/10.7554/eLife.59306.sa1
Author response https://doi.org/10.7554/eLife.59306.sa2

## Additional files

### Supplementary files

• Transparent reporting form

### Data availability

All data generated or analysed during this study are included in the manuscript and supporting files.

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

# Appendix 1

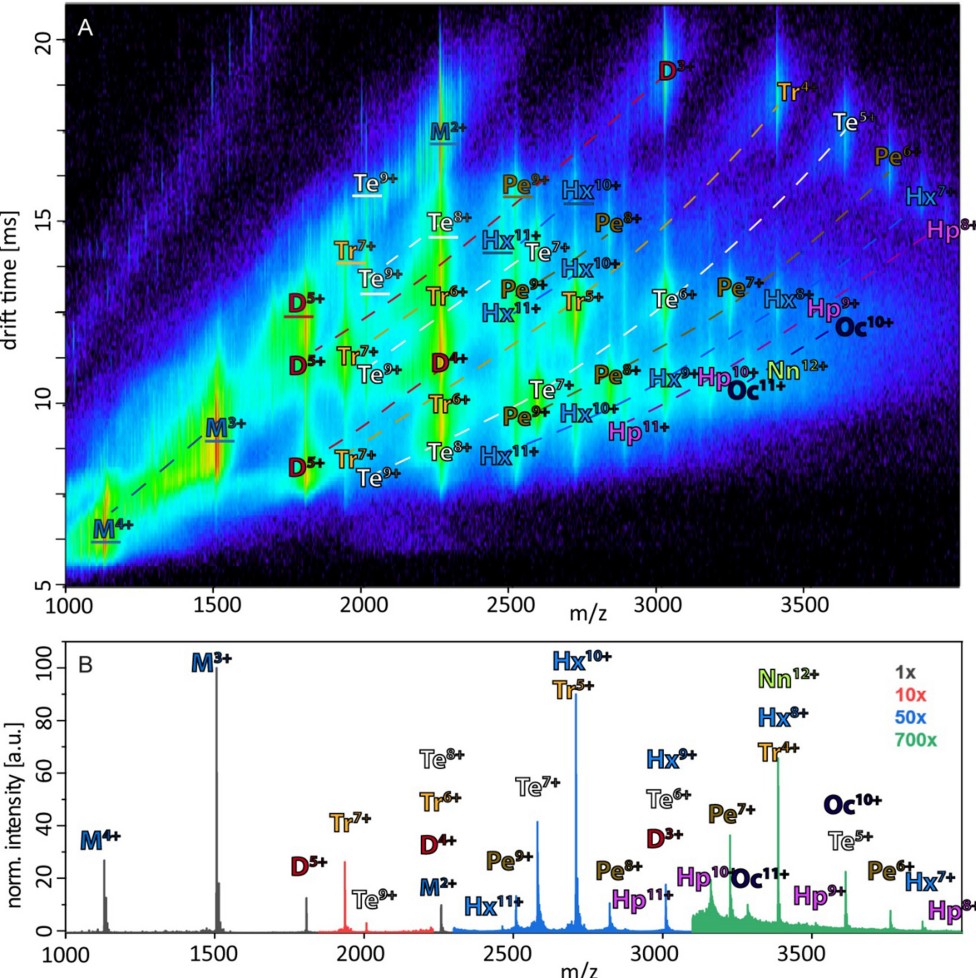

**Appendix 1—figure 1.** Annotated driftscope spectrum and corresponding mass spectrum. (**A**) Annotation of the driftscope spectrum of aggregated Aβ$_{42}$, shown in *Figure 1A*. The corresponding mass spectrum is shown in (**B**). Structured proteins appear in a driftscope plot already at lowest CE. Increase of CE leads to decrease of some of these structures, while new extended structures appear (See as well *Figures 2A* and *4C*, *Appendix 1—figure 5*). The driftscope in A is measured at a CE of 15V. This is sufficient for the higher charge states of many oligomers to unfold in the gas phase, as seen by the higher drift times (Unfolded oligomers are indicated by a white outline of the annotation, as opposed to the black outlined annotations, which depict those signals, which stem from solution structures). While many peaks overlap in the mass spectrum as well as the driftscope, unambiguous assignment of all species is possible via different means. In some cases identification is based on isotopic resolution (for examples see insets in *Figure 1A*), unique m/z ratio (such as M$^{3+}$, D$^{5+}$, Tr$^{7+}$,Te$^{7+}$ etc.) or as they are CIU products of a clearly assigned species. As different charge states of the same oligomeric species fall on characteristic curves in the driftscope (indicated by dashed lines) additional peaks can be assigned, if the other species of this series are already identified.

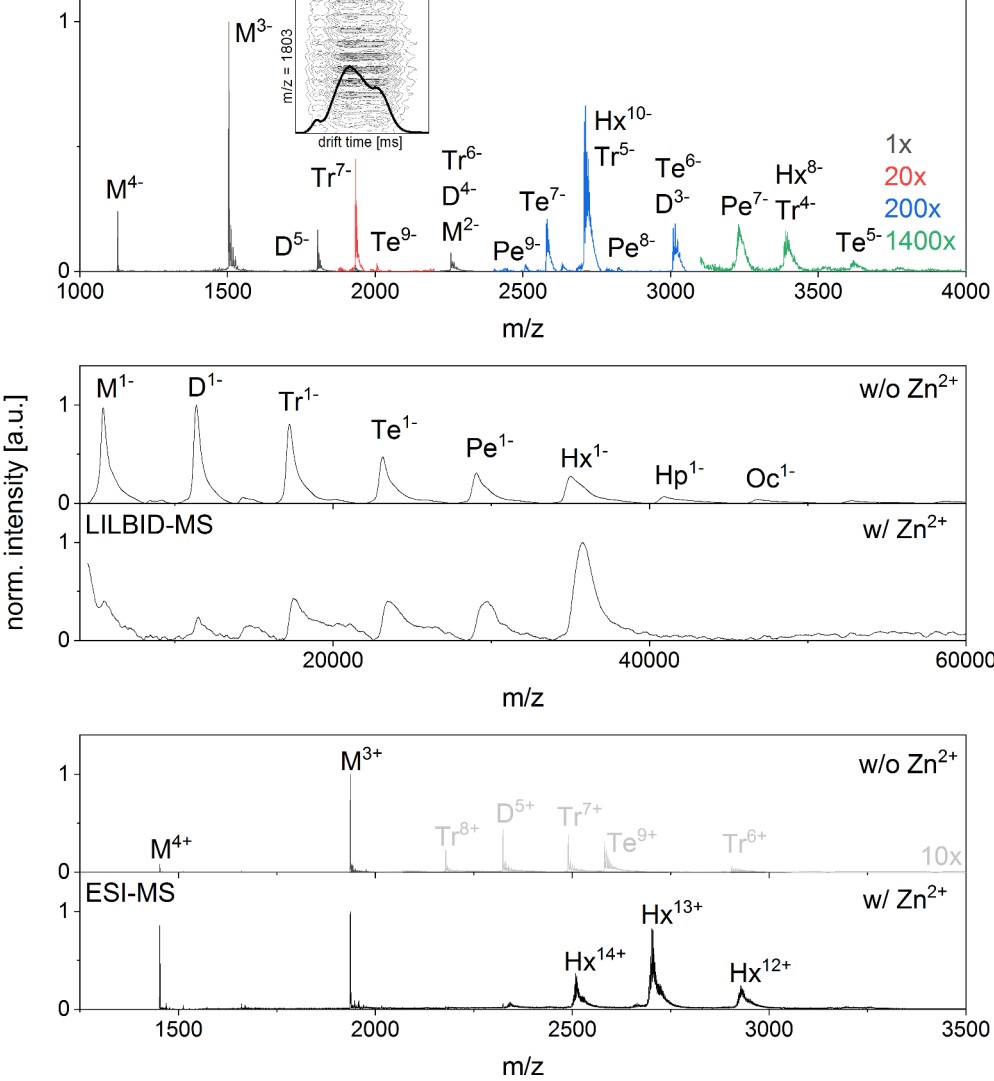

**Appendix 1—figure 2.** nESI mass spectrum of Aβ$_{42}$ in negative ion mode and LILBID and nESI spectrum of insulin showing a clear formation of a hexamer upon zink addition. (**A**) nESI Spectra of Aβ$_{42}$ in negative mode show similar characteristics to those taken in positive mode (compare *Appendix 1—figure 1*). Several oligomerization states can be seen, albeit at lower and lower intensity for the higher the oligomeric states (upscaling of the signal intensity was done as indicated in the figure). As the oligomerization of Aβ$_{42}$ was previously described as a process progressing via hexamers (*Bernstein et al., 2005*), we briefly explain here, why we support the notion of linear growth. Both MS methods applied here show Aβ$_{42}$ in different oligomeric states with decreasing intensity for higher oligomeric state and no mass peaks indicating a dominant hexamer. Mass spectra published by Bowers et al. don't show any dominant hexamers either, but the presence of tetramers and more dominant hexamers is deduced from different features seen in drift scope plots for the same m/z signals (*Bernstein et al., 2005*). Different drift times for one m/z ratio can be caused by different conformations of one oligomer species or by overlap of different charge states of different oligomers. Therefore for example the three different drift times for a nominal −5/2 charged monomer were interpreted as five times charged dimer, 10 times charged tetramer and 15 times charged hexamer. We reproduced these measurements and could similarly observe different drift times for m/z = 1806 in positive (*Figure 2B*) and negative ion mode (Inset in A). As our mass spectra show isotopic resolution we can nevertheless show that the alternative interpretation holds true: these drift times belong to different confirmations of only one oligomeric state – the dimer. This is true for all the species discussed in the work of Bowers and coworkers (*Bernstein et al., 2005*). We therefore see no indication for tetramer or hexamer in the drift time plots of these mass

peaks, and only very low intensity higher oligomers in the mass spectra. For comparison we investigated insulin, a protein, which is known to aggregate to a hexamer in the presence of zinc ions (*Dodson et al., 1979*) B shows LILBID results and C shows nESI results of insulin (10 µM insulin in 50 mM ammonium acetate buffer), both without (i) and with (ii) 1.5-fold excess of zinc respectively. Without zinc some aggregation can be seen with both methods, albeit more of the higher oligomers are retained in the LILBID spectrum, which is similar to our observations for Aβ$_{42}$ (compare *Appendix 1—figure 1* and *9*). Upon addition of zinc the oligomer distribution shifts to a dominant hexamer, which is clearly visible in both mass spectra. In contrast the hexamer appears in low intensity only for Aβ$_{42}$.

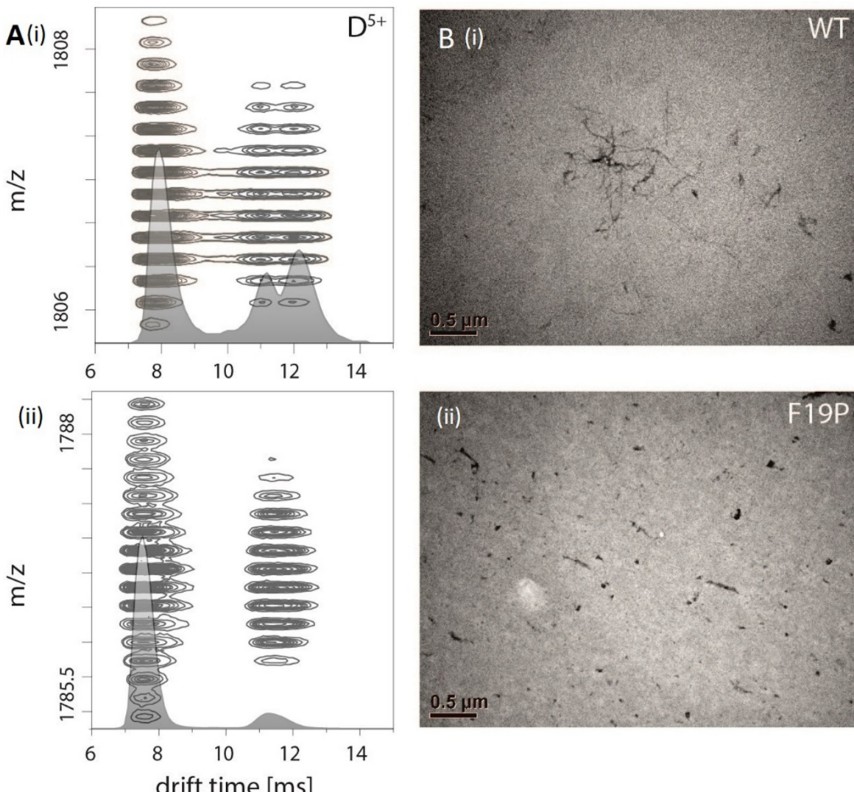

**Appendix 1—figure 3.** Comparision of Aβ$_{42}$ wt and mutant F19P. (**A**) Comparison of the IM spectrum of the 5-times charged dimeric wildtype Aβ$_{42}$ (**i**) with that of the Aβ$_{42}$ mutant F19P (**ii**) shows loss of the DB structure for the later one. (**B**) TEM images of both species show that this goes along with the loss of the characteristic fibrillary structures, which can only be seen for Aβ$_{42}$ wt.

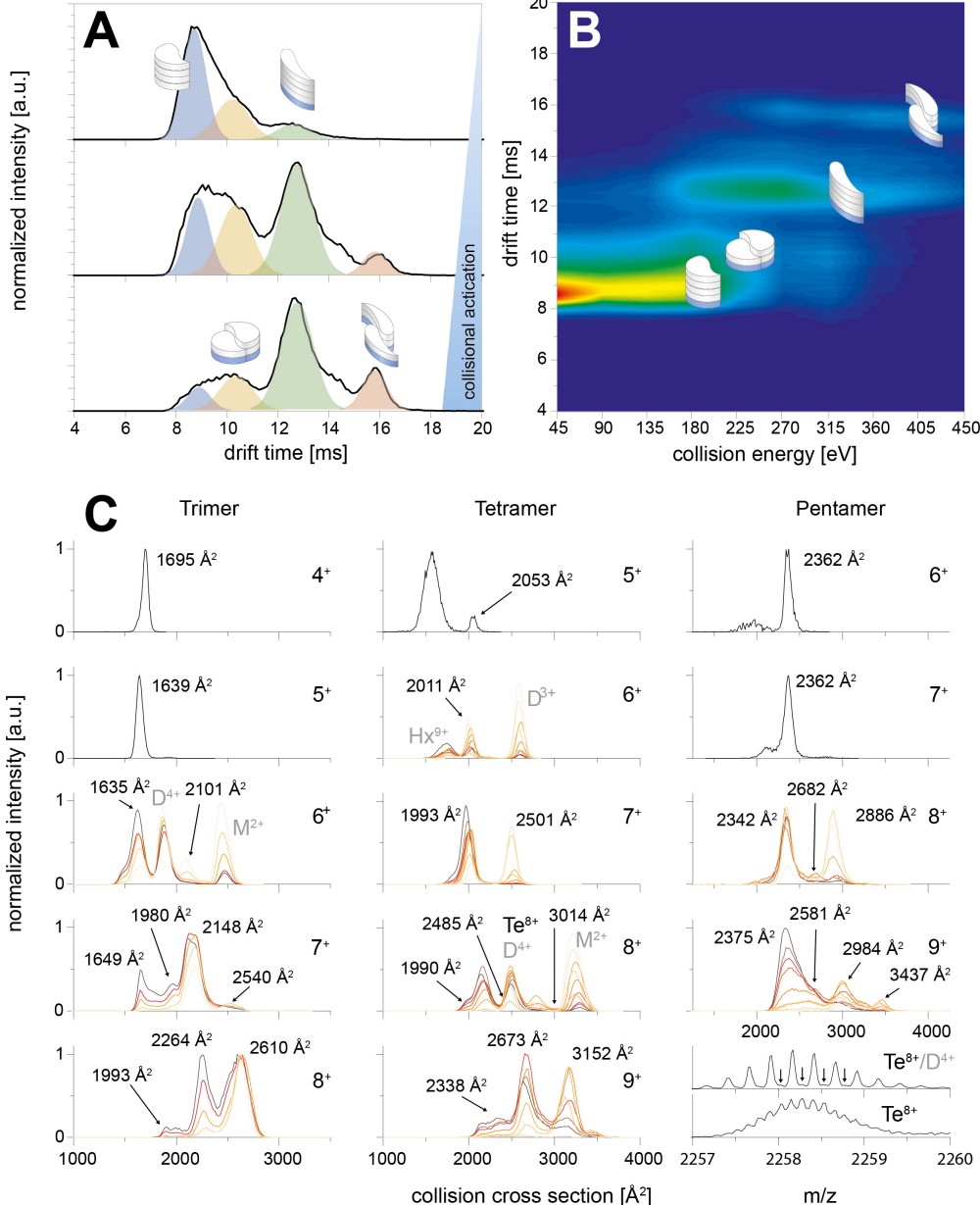

**Appendix 1—figure 4.** CIU of the trimer, tetramer and pentamer species. The schemes indicate the different species (folded and unfolded MB and DB structure). A shows three IM spectra for different CEs. In B the spectra of the whole CIU experiment are shown as heat map and also correlated to the four structures (folded and unfolded MB and DB). Similarly C shows IM spectra for trimer, tetramer and pentamer for different charge states. The spectra are overlaid for different CE. In all cases the smaller conformers pick up less charges in the ESI process, allowing to observe the two conformations (MB and DB) only for medium and higher charge states. For the lowest charge states the lab frame energies achieved with CE ramping are not sufficient for CIU. MB oligomers with medium charge states undergo unfolding prior to CID. The collisional energy experienced by the oligomers of higher charge states leads to unfolding and then dissociation for DB and MB species. The CCS values of the relevant species are indicated in the respective plots; where other species overlap (such as the dimer and tetramer species in the 4-times charged dimer spectra) they are indicated as well.

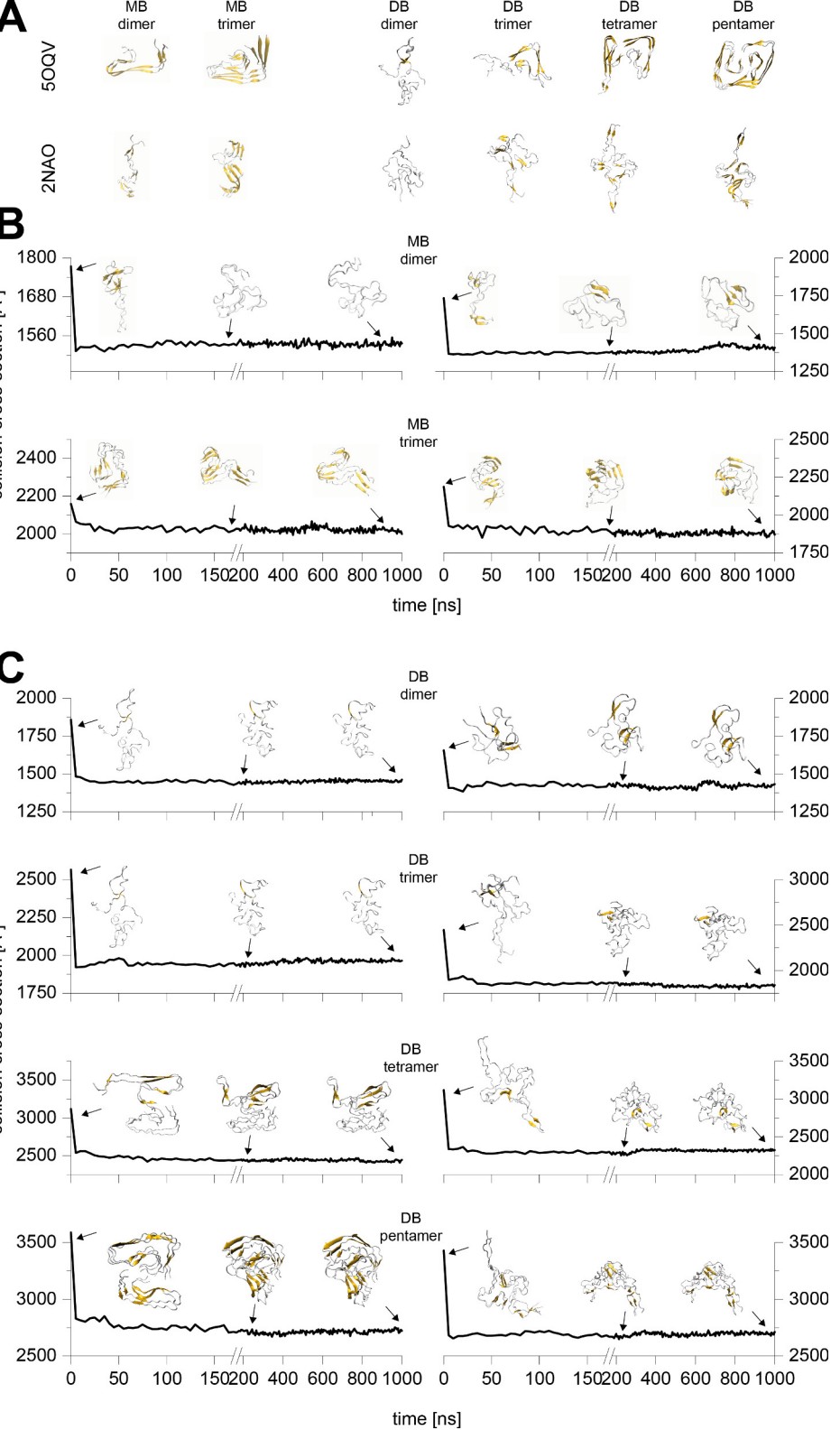

**Appendix 1—figure 5.** Typical results from the MD simulations of different Aβ$_{42}$ oligomers cropped out of the PDB structures 5OQV and 2NAO. A shows the respective structure after 10 ns simulation in water. These structures were sent to vacuum simulations. B and C depict the change of the structure's CCS during simulation in vacuum for 1 µs. The insets show exemplary structures after 0 ns, 200 ns and 1000 ns of simulation. The CCS of the 200 and 1000 ns vacuum structures differ by

2% at most, which is below the resolution of the IM-MS. Hence all the remaining calculations were performed to 200 ns.

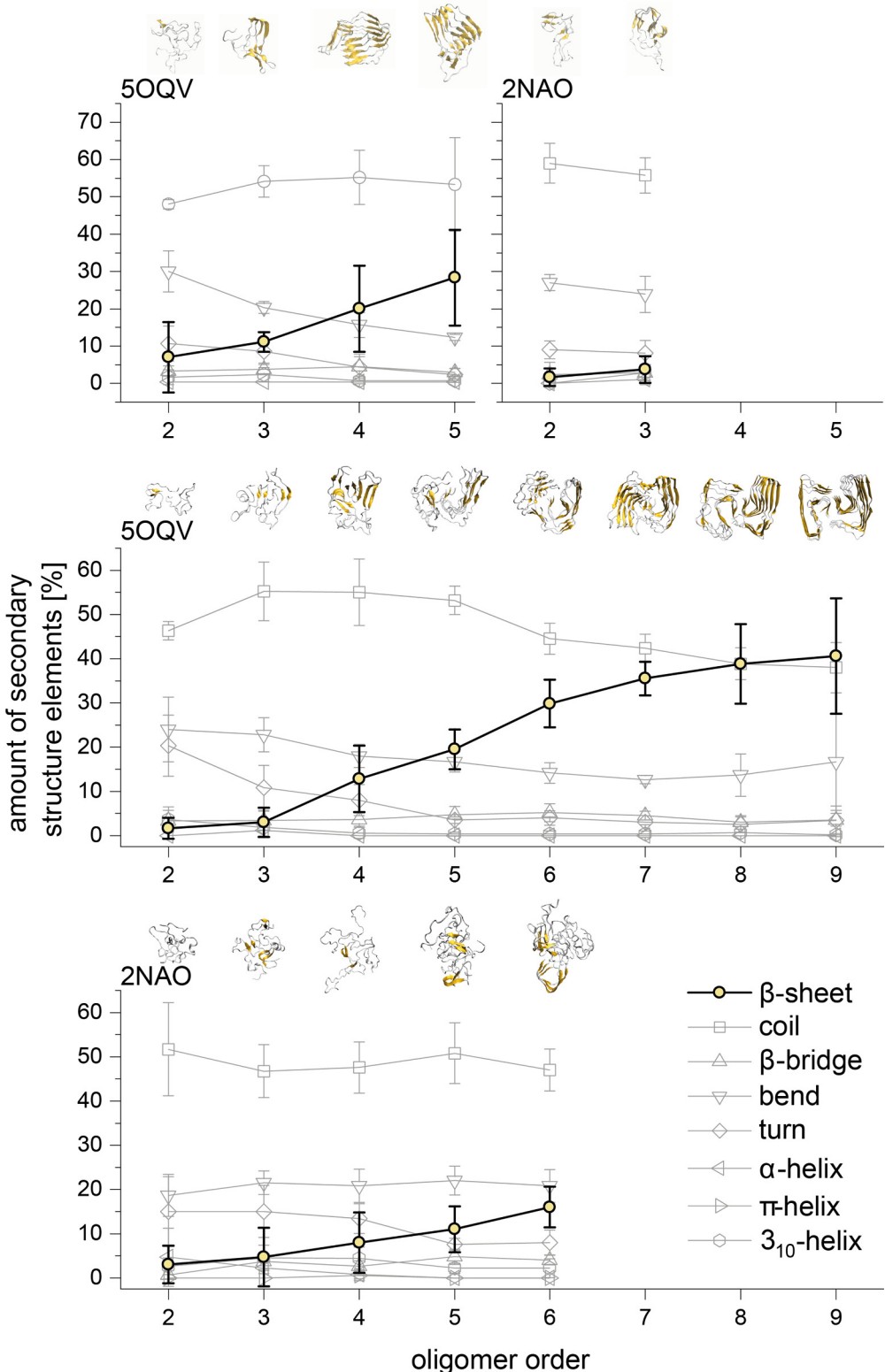

**Appendix 1—figure 6.** Development of the different structural elements of the Aβ$_{42}$ oligomers as calculated using the DSSP algorithm for MB oligomers (top) and DB oligomers (middle and bottom). In all cases the most striking feature is the development of the β-sheet motive for the higher

oligomers. The structures above show these motives in gold for the different representative Aβ42 oligomers.

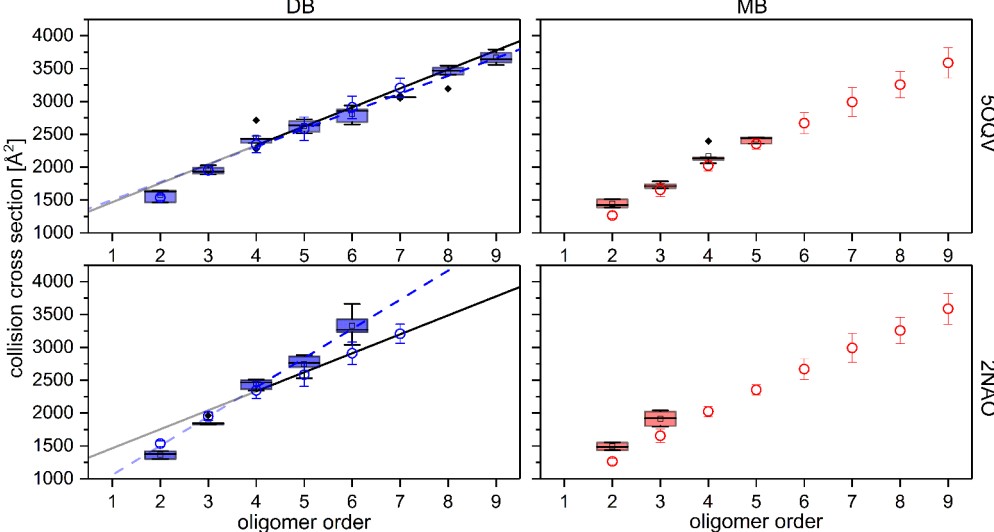

**Appendix 1—figure 7.** Comparison of experimental CCS values (blue circles for DB conformation on the left and red circles for MB conformation on the right) to CCS values of MD simulated PDB structures (box plots, using a minimum of five calculated structures) of Aβ42 oligomers. For comparison the structures detected with cryo-EM by Gremer, et al. (PDB 5OQV) and with solid-state NMR by Wälti, et al. (PDB 2NAO) were used for MD simulations and CCS calculations as illustrated above (*Gremer et al., 2017*; *Wälti et al., 2016*). Linear fits for oligomers from tetramer on, demonstrate the difference between experimental (solid) and theoretical (dashed) CCS of the DB conformation. The structure of PDB 5OQV (upper plots) matches our experimentally determined CCS values best.

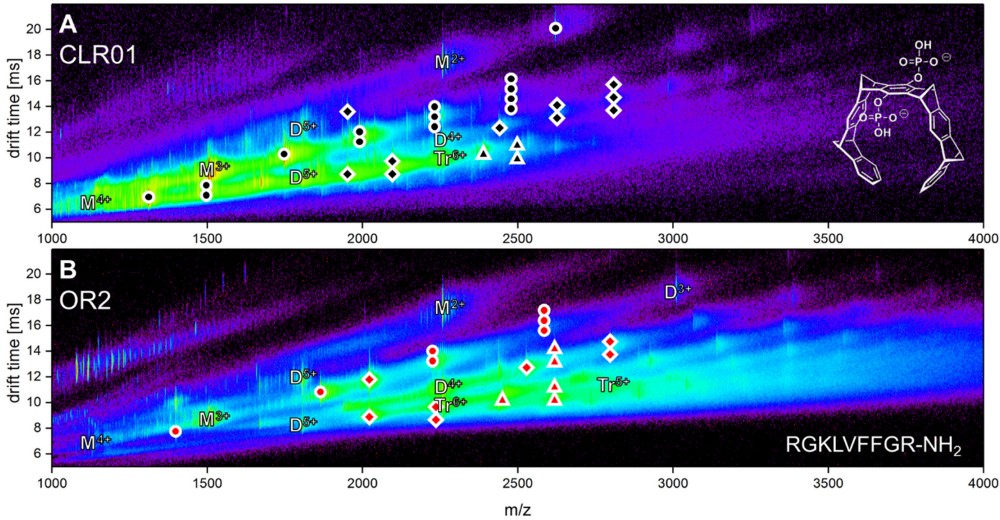

**Appendix 1—figure 8.** Driftscope of Aβ42 interacting with the two ligands CLR01 (**A**) and OR2 (**B**). The marker indicates an interaction of an Aβ42 oligomer with up to four copies of the respective molecule. Higher binding events are not indicated (ligand interaction with: circle = monomer, square = dimer, triangle = trimer).

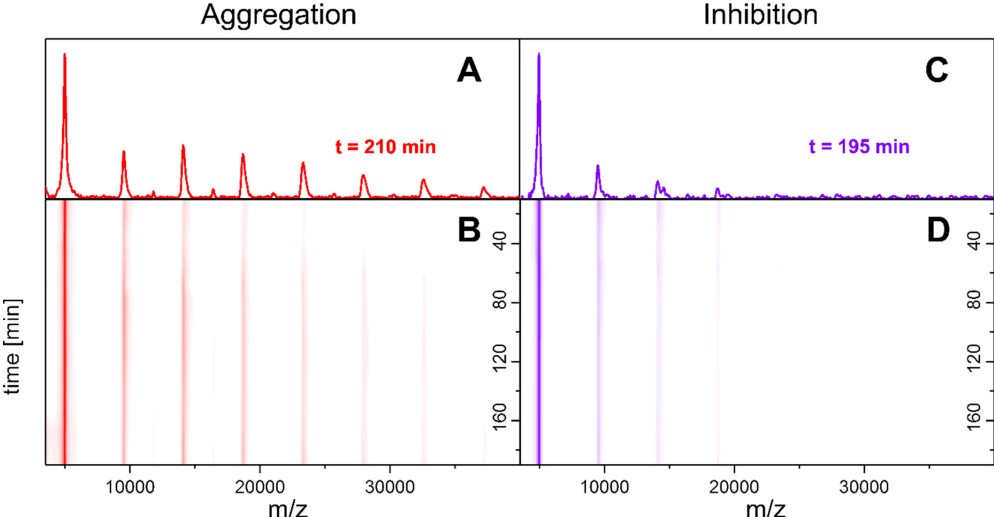

**Appendix 1—figure 9.** Time-resolved measurements of the oligomerization of Aβ$_{42}$ incubated at 22° C. LILBID-MS detects the development of the first oligomers (**A, B**), which is hindered by the presence of CLR01 (**C, D**). A and C depict spectra recorded after 200 ± 10 min of Aβ$_{42}$ incubation in the absence and presence of CLR01 respectively. B and D show the time-course of the appearance of Aβ$_{42}$ oligomers. Without CLR01 (**B**) oligomers increase in size. CLR01 inhibits this aggregation (**D**).

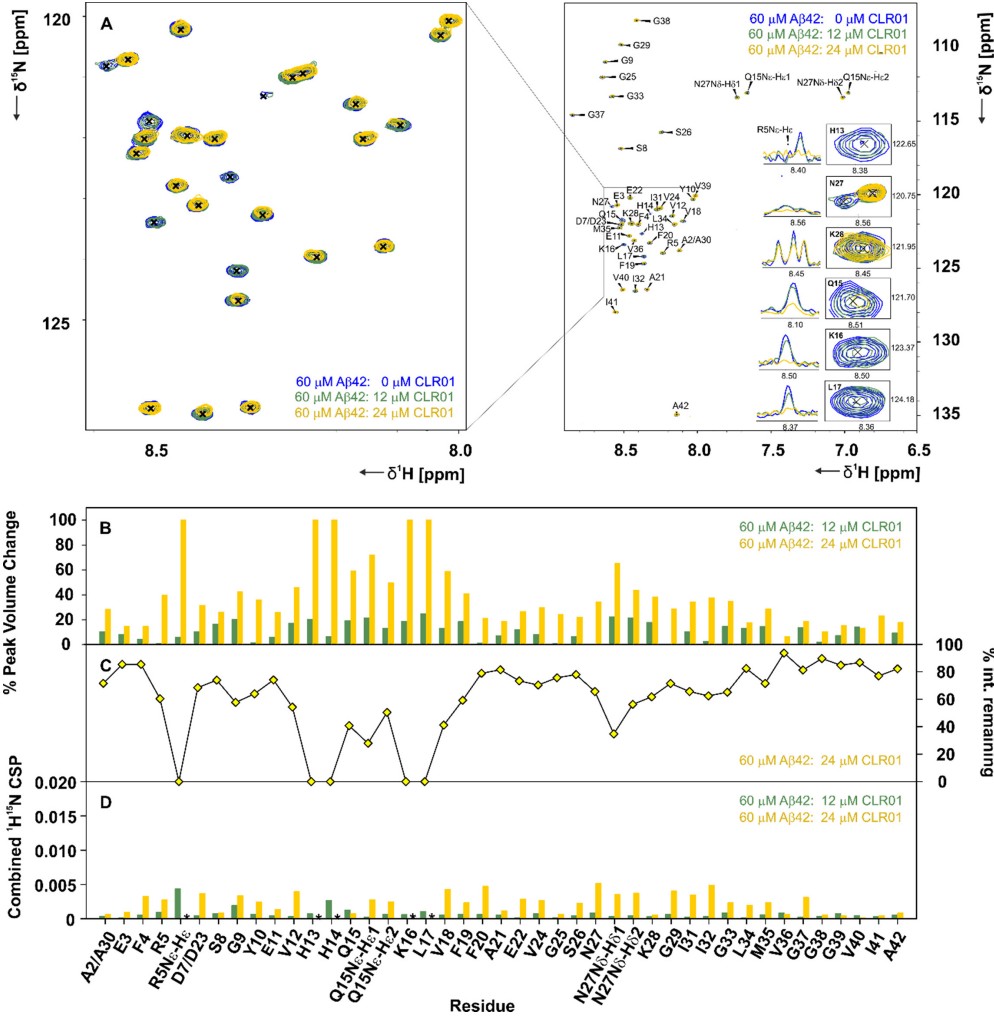

**Appendix 1—figure 10.** CLR01 interacts with Aβ$_{42}$in specific areas. (**A**) A 2D-{$^1$H, $^{15}$N}-HSQC overlay of 60 μM Aβ$_{42}$ measured at 600 MHz and 278 K in the presence of 12 μM and 24 μM CLR01, depicted in green and yellow respectively. The spectra were recorded at 600 MHz and 278.5 K in 15 mM sodium phosphate, 55 mM NaCl, pH 7.2, 10% D$_2$O. An external reference containing 0.3 mM DSS was used for calibration. Inset (**A**), enlarged on the left, depicts the 2D-{$^1$H, $^{15}$N}-HSQC overlay focusing on the interacting region, including residues H13, Q15, K16, L17, N27, and K28, and the respective 1D-$^1$H projections of a set of 2D-{$^1$H, $^{15}$N}-HSQC peaks of Aβ$_{42}$ after addition of 0 μL, 12 μL and 24 μL CLR01. (**B**) Illustrates the % peak volume change of residues of Aβ$_{42}$ in the absence of CLR01, versus Aβ$_{42}$ with 12 μM and 24 μM CLR01 respectively. (**C**) The % intensity remaining for 60 μM Aβ$_{42}$: 24 μM CLR01 versus 60 μM Aβ$_{42}$ plotted according to residue. (**D**) Spectral differences were mapped in the CSP chart according to the equation:CSP=((0.1·δ$^{15}$N$_{ref}$-0.1·δ$^{15}$N)$^2$+(δ$^1$H$_{ref}$-δ$^1$H)$^2$)$^{0.5}$. The combined chemical shift perturbation is shown for 60 μM Aβ$_{42}$ alone versus 60 μM Aβ$_{42}$: 12 μM and 24 μM CLR01 respectively. CSPs not depicted due to the disappearance of peaks are denoted by an asterisk.

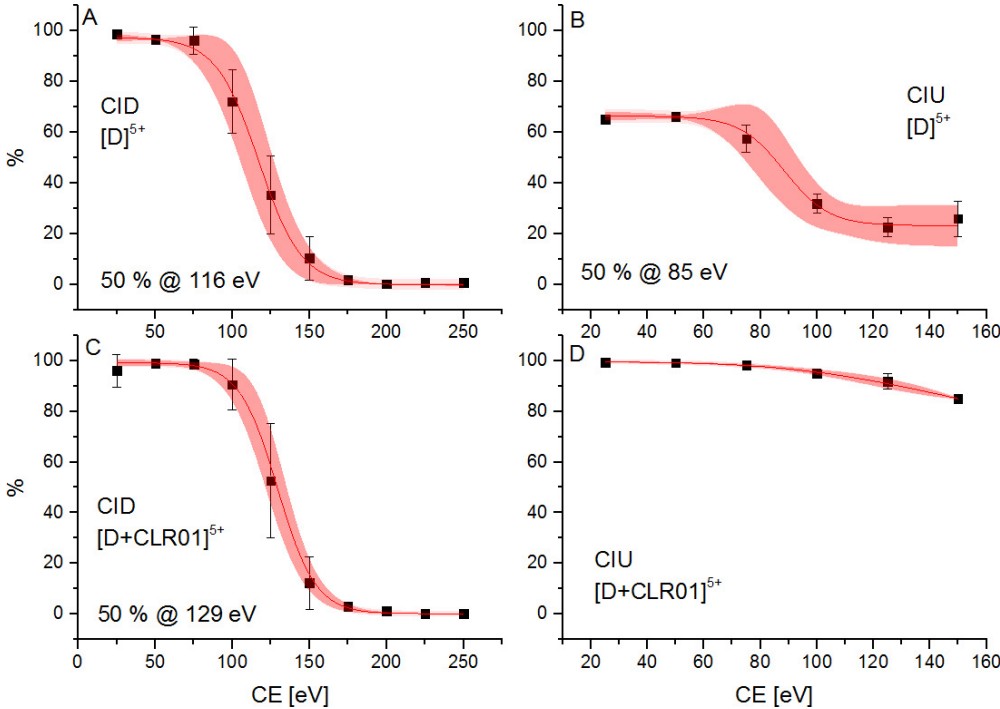

**Appendix 1—figure 11.** Effect of ramping the CE on CID and CIU events of the 5-times charged dimer of free Aβ$_{42}$ (**A and B**) and CLR01 bound Aβ$_{42}$ (**C and D**). The energies necessary to dissociate or unfold 50% of the species are given, where applicable. In the case of CLR01 bound Aβ$_{42}$ 50% unfolding is not achieved prior to dissociation. The areas shaded in red represent the 90% confidence region of three consecutive measurements.

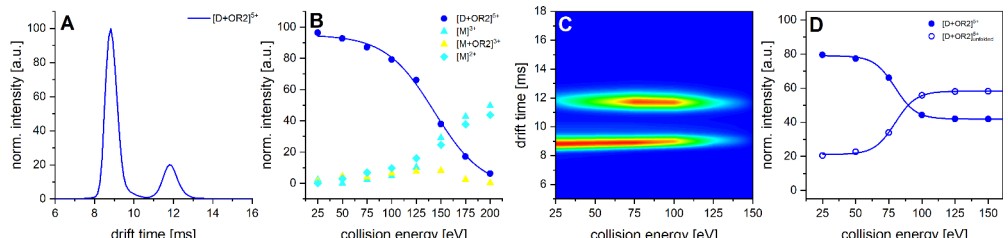

**Appendix 1—figure 12.** CID and CIU experiments of Aβ$_{42}$ with OR2. A shows the IM spectrum of the 5-times charged dimer of OR2 bound Aβ$_{42}$. The signal of the folded and unfolded MB species are visible, while no signal can be observed for a DB conformation. B shows the observed CID in dependence of CE increase. C depicts a heat map of the CIU experiment of the 5-times charged dimer peak of OR2 bound Aβ$_{42}$. D shows an intensity plot of the peaks visible in the CIU experiment of B.

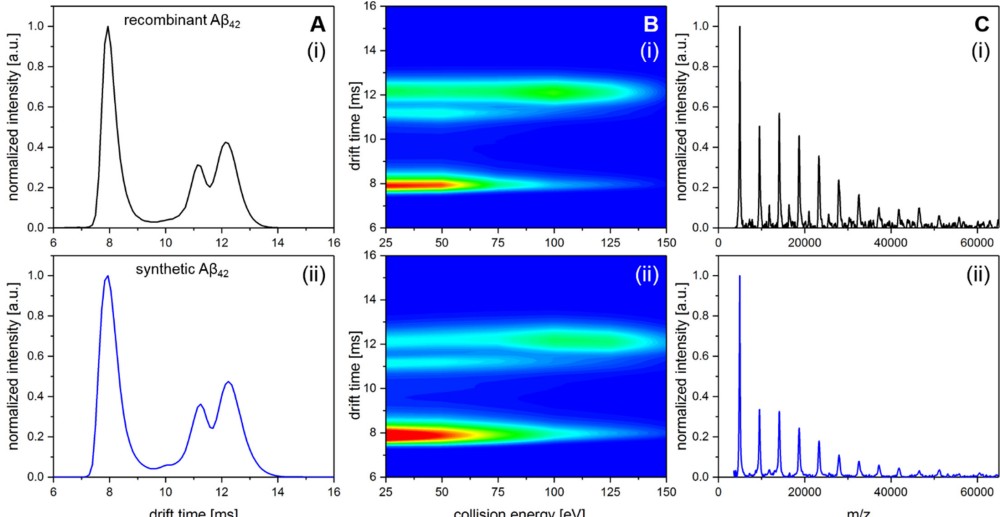

**Appendix 1—figure 13.** Comparison of recombinant Aβ₄₂ and Aβ₄₂ synthesized via Fmoc chemistry. A shows IM spectra of the $D^{5+}$ signal for recombinant (i) and synthetic (ii) peptide. In both cases all signals in the respective IM spectra appear at the identical drift times. B shows heat maps of the CIU experiment for recombinant (i) and synthetic (ii) peptide. Only a slight difference in signal intensity between the compact and the unfolded MB dimer can be observed. However, the ratio where 50% of dimeric Aβ₄₂ is unfolded appears in both cases to be at about 85 eV. C shows LILBID spectra for recombinant (i) and synthetic (ii) peptide incubated at 22°C for 200 min. Intensity differences of the oligomers between (i) and (ii) are on the level of reproducibility of the experiments.

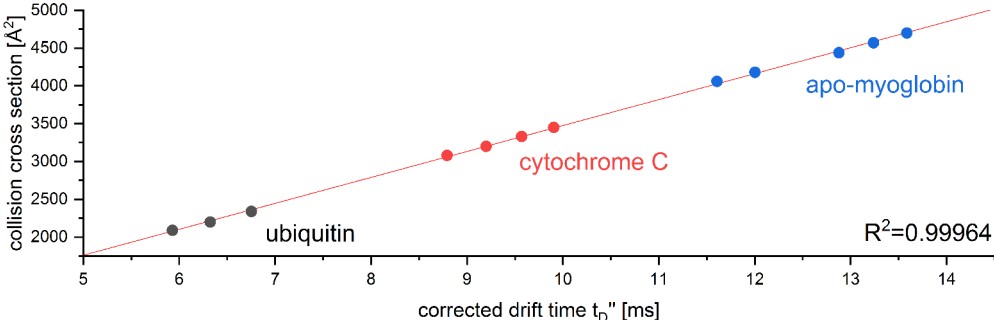

**Appendix 1—figure 14.** CCS calibration curve for the reference proteins used to determine the CCS of Aβ₄₂ oligomers in positive ion mode. The calibration was performed under denaturing conditions according to *Ruotolo et al., 2008*. The corrected drift time was plotted against the reference CCS obtained from Bush Lab CCS database.

