## [Decision Letter]

**Acceptance summary:**

The insights that this paper provides into the oligomerization mechanisms of the Amyloid-β_42_ peptide represent a new understanding of the distinction between pathways that leads to toxic fibrils, and those that lead to amorphous, and non-toxic, aggregation. The finding that the inhibitor CLR01 inhibits the toxic pathway will be important for future drug development efforts for Alzheimer's disease.

**Decision letter after peer review:**

Thank you for submitting your article "Structural Rearrangement of Amyloid-β upon Inhibitor Binding Suppresses Formation of Alzheimer Disease Related Oligomers" for consideration by *eLife*. Your article has been reviewed very favorably by two peer reviewers, and the evaluation has been overseen by John Kuriyan as the Reviewing Editor. The following individual involved in review of your submission has agreed to reveal their identity: Hugh I Kim (Reviewer #2).

The reviewers have discussed the reviews with one another and the Reviewing Editor has drafted this decision to help you prepare a revised submission. Note that the reviewers are only asking for minor changes to clarify matters under discussion, and that no further experimental analysis is called for. At *eLife* it is our normal practice to provide authors with a consolidated review in which the points made by the reviewers are merged into one set of recommendations. In this case, because the points raised are straightforward, we are providing you with the separate reviews.

Reviewer #1:

The manuscript describes an approach to characterize Aβ_42_ in the early stages of the aggregation pathway. By analysing smaller oligomers via IM-MS and estimating the growth of both species based on their collision cross-sections, the authors identify two aggregation pathways. While the toxic aggregation pathway starts with the stacking of Aβ_42_ dimers (DB), the second, non-toxic aggregation pathway is based on the stacking of Aβ_42_ monomers (MB). This assignment is supported by earlier studies based on solid-state NMR and cryo-EM. Understanding the structural properties of both aggregation pathways allows the authors to draw conclusions about the toxicity of different oligomers and to determine the interaction modes of two different inhibitors with Aβ_42_.

The lack of suitable analytical methods to study the amyloid assembly of Aβ_42_ is a tremendous challenge and the proposed approach is therefore of broad interest. The presented results are promising, especially the match between experimental and theoretical CCSs of the TEM based structures is surprisingly good. This study could be an important step to rapidly monitor the aggregation pathway of Aβ_42_ and identify potential drug candidates.

The manuscript is generally suited to be published in *eLife*. There are a few points that should be addressed prior to publication:

1) Figure 1B indicates, that both species (MB and DB) follow fibrillar growth. However, according to the introduction of the manuscript (and to literature) MB is known to form amorphous structures, for which an isotropic growth is expected. Can the authors explain this discrepancy?

2) The fitting of isotropic/linear growth for MB should be discussed in more detail. While linear growth is very clear for DB (starting from the tetramer), the fit of MB could be both linear and isotropic within the given error. This aspect should be discussed in more detail as the growth behaviour of MB and DB is the basis for all further interpretation.

3) The text would benefit from a rearrangement, shortening, and simplification of some paragraphs. Especially subsection “CLR01 inhibits A*β*_42_ oligomerization” explaining the inhibition of Aβ_42_ is hard to follow for the reader.

Reviewer #2:

This manuscript by Lieblein et al. demonstrated the Aβ_42_ oligomerization mechanisms in the early stage of fibrillation using native mass combined with ion mobility mass spectrometry (IM-MS). The authors observed Aβ_42_ oligomers from monomer to nonamer with various charge states in IM-MS spectra and stated that Aβ_42_ oligomers develop into two distinct arrangements leading to protein aggregates. The authors also investigated the inhibition effects of ligand and peptide on Aβ_42_ fibrillation based on ionic and hydrophobic interactions using CID and CIU experiments. Several IM-MS studies had been exploring the oligomerization process of amyloidogenic protein, but Lieblein et al. propose the fundamental and novel mechanism of fibril growth of Aβ_42_ using MS-based techniques. Therefore, I recommend this manuscript for publication after minor revision. Below are specific comments to the manuscript.

1) Subsection “CLR01 inhibits A*β*_42_ oligomerization”: The authors selected two inhibitors, OR2 and CRL01, to investigate the inhibitory effects of the aggregation of Aβ_42_ oligomers by different aggregation pathways. However, the authors described the inhibitory effect of CLR01 only by performing time-resolved LILBID-MS, indicating the adherence of small oligomers in the presence of CLR01, without describing the OR2 effect. If this section does not explain the effect of OR2 on suppressing Aβ_42_ oligomerization, it would be better not to mention OR2 in the second sentence of this paragraph.

2) Subsection “Oligomer stability is size-dependent”: The author should add more CIU graphs for all the oligomers, for the explanation "For all oligomers the higher charge states show unfolding for both species"

3) Subsection “MB and DB structures”: The authors should provide MD simulation figures for the MB cases, like Appendix 1—figure 5 and 6.

4) Subsection “Inhibitors stop origin of dimer-based oligomers”: The authors should add CID data for CE need for 50% CID for OR2 for the explanation "Both ligands increase the CE needed for 50% CID slightly".

---

## [Author Response]

Reviewer #1:The manuscript describes an approach to characterize Aβ_42_ in the early stages of the aggregation pathway. By analysing smaller oligomers via IM-MS and estimating the growth of both species based on their collision cross-sections, the authors identify two aggregation pathways. While the toxic aggregation pathway starts with the stacking of Aβ_42_ dimers (DB), the second, non-toxic aggregation pathway is based on the stacking of Aβ_42_ monomers (MB). This assignment is supported by earlier studies based on solid-state NMR and cryo-EM. Understanding the structural properties of both aggregation pathways allows the authors to draw conclusions about the toxicity of different oligomers and to determine the interaction modes of two different inhibitors with Aβ_42_.The lack of suitable analytical methods to study the amyloid assembly of Aβ_42_ is a tremendous challenge and the proposed approach is therefore of broad interest. The presented results are promising, especially the match between experimental and theoretical CCSs of the TEM based structures is surprisingly good. This study could be an important step to rapidly monitor the aggregation pathway of Aβ_42_ and identify potential drug candidates.The manuscript is generally suited to be published in eLife. There are a few points that should be addressed prior to publication:1) Figure 1B indicates, that both species (MB and DB) follow fibrillar growth. However, according to the introduction of the manuscript (and to literature) MB is known to form amorphous structures, for which an isotropic growth is expected. Can the authors explain this discrepancy?2) The fitting of isotropic/linear growth for MB should be discussed in more detail. While linear growth is very clear for DB (starting from the tetramer), the fit of MB could be both linear and isotropic within the given error. This aspect should be discussed in more detail as the growth behaviour of MB and DB is the basis for all further interpretation.

To (1) and (2) The reviewer is correct. The MB species will form amorphous structures, which should follow our isotropic growth model. The error bars for the data points for the relevant higher MB oligomers do not allow to exclude either of the models. We described this now more clearly in the manuscript.

3) The text would benefit from a rearrangement, shortening, and simplification of some paragraphs. Especially subsection “CLR01 inhibits Aβ_42_ oligomerization” explaining the inhibition of Aβ_42_ is hard to follow for the reader.

We tried to improve the respective explanations and the readability.

Reviewer #2:This manuscript by Lieblein et al. demonstrated the Aβ_42_ oligomerization mechanisms in the early stage of fibrillation using native mass combined with ion mobility mass spectrometry (IM-MS). The authors observed Aβ_42_ oligomers from monomer to nonamer with various charge states in IM-MS spectra and stated that Aβ_42_ oligomers develop into two distinct arrangements leading to protein aggregates. The authors also investigated the inhibition effects of ligand and peptide on Aβ_42_ fibrillation based on ionic and hydrophobic interactions using CID and CIU experiments. Several IM-MS studies had been exploring the oligomerization process of amyloidogenic protein, but Lieblein et al. propose the fundamental and novel mechanism of fibril growth of Aβ_42_ using MS-based techniques. Therefore, I recommend this manuscript for publication after minor revision. Below are specific comments to the manuscript.1) Subsection “CLR01 inhibits Aβ_42_ oligomerization”: The authors selected two inhibitors, OR2 and CRL01, to investigate the inhibitory effects of the aggregation of Aβ_42_ oligomers by different aggregation pathways. However, the authors described the inhibitory effect of CLR01 only by performing time-resolved LILBID-MS, indicating the adherence of small oligomers in the presence of CLR01, without describing the OR2 effect. If this section does not explain the effect of OR2 on suppressing Aβ_42_ oligomerization, it would be better not to mention OR2 in the second sentence of this paragraph.

The reviewer is correct. We removed the respective mentioning of OR2.

2) Subsection “Oligomer stability is size-dependent”: The author should add more CIU graphs for all the oligomers, for the explanation "For all oligomers the higher charge states show unfolding for both species"

We added the requested CIU graphs in Appendix 1—figure 4. They show the additional unfolded DB structure for the higher charge states.

3) Subsection “MB and DB structures”: The authors should provide MD simulation figures for the MB cases, like Appendix 1—figure 5 and 6.

We added the requested plots into Appendix 1—figure 5 and 6.

4) Subsection “Inhibitors stop origin of dimer-based oligomers”: The authors should add CID data for CE need for 50% CID for OR2 for the explanation "Both ligands increase the CE needed for 50% CID slightly".

We added the graph showing the CID experiment for OR2 in Figure S12 (B).